# Ontogeny shapes individual dietary specialization in female European brown bears (*Ursus arctos*)

Anne G. Hertel [1,2] ✉, Jörg Albrecht [2], Nuria Selva [3,4,5], Agnieszka Sergiel [3], Keith A. Hobson [6,7,15], David M. Janz [8], Andreas Mulch [2,9], Jonas Kindberg [10,11], Jennifer E. Hansen [12], Shane C. Frank [12], Andreas Zedrosser [12,13] & Thomas Mueller [2,14]

Individual dietary specialization, where individuals occupy a subset of a population's wider dietary niche, is a key factor determining a species resilience against environmental change. However, the ontogeny of individual specialization, as well as associated underlying social learning, genetic, and environmental drivers, remain poorly understood. Using a multigenerational dataset of female European brown bears (*Ursus arctos*) followed since birth, we discerned the relative contributions of environmental similarity, genetic heritability, maternal effects, and offspring social learning from the mother to individual specialization. Individual specialization accounted for 43% of phenotypic variation and spanned half a trophic position, with individual diets ranging from omnivorous to carnivorous. The main determinants of dietary specialization were social learning during rearing (13%), environmental similarity (5%), maternal effects (11%), and permanent between-individual effects (9%), whereas the contribution of genetic heritability (3%) was negligible. The trophic position of offspring closely resembled the trophic position of their mothers during the first 3–4 years of independence, but waned with increasing time since separation. Our study shows that social learning and maternal effects were more important for individual dietary specialization than environmental composition. We propose a tighter integration of social effects into studies of range expansion and habitat selection under global change.

Among individuals of the same species, ecological niche variation is common and may occur when the availability of food resources or habitat structure changes across the species' range. Individual variation is key for driving species resilience in response to shifting resource availabilities in a rapidly changing world, and may ultimately determine local persistence or extinction of species[1]. Ecological generalists, species with a wide ecological niche, also seem to exhibit more individual specialization (i.e. between-individual variation of niche)[2] and are likely particularly well adapted to persist under shifts in resource availability or composition, enabling them to occupy larger distributional ranges than ecological specialists[3]. Inter- and intraspecific competition, predation, and ecological opportunity alter resource availability and have been identified as the main ecological drivers explaining variation in the degree of individual dietary specialization among populations[4]. However, how individual variation in dietary specialization emerges and is maintained within populations has, to our knowledge, not been quantified in the wild.

In the fields of behavioral and evolutionary biology, individual variation is measured as the variance attributed to permanent between-individual differences, while the sources of variation can be

quantified using complex hierarchical models (e.g., "animal model")[5]. In principle, three sources of variation are commonly considered[5,6]: variation in the environment[7], additive genetic effects from which trait heritability can be estimated[8,9], and parental (especially maternal) effects[10,11]. In addition, individual variation can be maintained through social learning during ontogeny[12], an aspect that, to our knowledge, has rarely been integrated into animal models (but see refs. 13,14). We here provide a study to attribute individual variation in the dietary specialization to its sources.

Differences in the environment, in terms of habitat composition and associated availability of particular food resources, are generally considered the main cause of individual variation in dietary specialization[15]. This is particularly true in range-resident species, where individuals occupy a subset of the population's range and individual home ranges vary in resource availability[16]. However, beyond the environment, resource preferences have been suggested to be genetically heritable and determined through genes inherited from both mother and father, where more closely related individuals share more similar diets than distantly related individuals[15,17]. Additionally, parental phenotypes may also affect offspring phenotypes in ways other than genetic heritability[10,11]. Maternal effects are more commonly studied because mothers often have unilateral control over offspring development[10], especially in mammals, however, paternal effects are plausible in species with paternal care. Maternal effects on offspring behavior have been suggested to be lifelong, they have either a genetic or environmental basis and summarize the cumulative influence of many different proximate maternal effects, including pathways such as provisioning rates, milk production, in-utero hormone transfer, and epigenetics[10]. Statistically, maternal effects account for similarities in dietary niche among offspring of the same mother (fitted as a random intercept for mother identity)[13,14], but not for the similarity of dietary niche between mother and offspring. The latter would be an example where the maternal trait affects the offspring's trait, which statistically can be clearly differentiated from other maternal effects[13,14]. Similarities between the dietary phenotypes of mothers and their offspring indicate social learning of resource preference or competence to secure a resource by the offspring from the mother during early ontogeny[18–22]. Social learning is therefore an additional pathway by which individual variation can be maintained. It is reasonable to assume that the effects of social learning during rearing will weaken later in life through individual-experiential learning[23].

Attributing variation in diet to the individual level, isolating its sources, and identifying developmental drivers of diet preferences in the wild requires multigenerational datasets of repeated measures of the diet of individuals throughout their life[5]. We used a unique 30-year longitudinal dataset of 71 female Scandinavian brown bears (*Ursus arctos*) of known mothers with repeated annual isotopic estimates of trophic position to study, the sources of individual dietary specialization in the wild. Brown bears are ecological generalists with a distribution range spanning the northern hemisphere from tundra to deserts, paralleled by extensive variation in diet. Populations range from tracking food resource pulses, such as spawning fish[24], scavenging on ungulate carcasses or preying on ungulate calves[25], or feeding extensively on invertebrates, to populations using primarily fruiting plant-based diets[26,27]. Given such extreme dietary plasticity, it is not surprising that great dietary variation has been found also within populations[28,29]; however, the determinants and ontogeny of this variation at the individual level remain largely unknown. In ecology, differences in diet are often primarily attributed to differences in resource availability and abundance. Even within populations inhabiting a continuous biome, home range scale variation in habitat composition[30] can lead to variation in resource availability. Brown bears maintain non-territorial home ranges and the most parsimonious source of individual specialization is, therefore, heterogeneity in the environment. It is further plausible that individual specialization in brown bears is genetically heritable. For example, while body size is determined largely by resource availability in the environment, it has also been shown to be genetically heritable in our study population[31], suggesting greater similarity among closely related individuals also in other linked traits, such as trophic position. In addition, maternal effects could shape individual specialization in brown bear offspring. As a potential pathway, milk quantity or quality[32] can vary among females due to genetic differences and/or differences in their home range quality, leading to consistently larger or smaller offspring from the same mother, which in turn could cause similarities in trophic position among siblings. Last, brown bears live a solitary lifestyle except for the period of offspring rearing involving up to three years of maternal care[33], after which female offspring often settle close to their mother's home range[34]. In their first years of life, bear cubs accompany their mother, so it is reasonable to predict that brown bear offspring learn their dietary niche from their mothers. If mothers differ in dietary niches, these differences may be maintained in the population through offspring social learning from the mother (hereafter "social learning").

Individual trophic position is one metric to assess individual specialization along a continuum from a more plant-based to a more meat- or insect-based diet. Trophic position can be estimated from the ratio of stable-nitrogen isotopes ($\delta^{15}N$) in growing tissue and reflects cumulative diet intake during the period of tissue growth. Individuals with higher trophic positions are specialized on more protein-rich diets, relative to individuals with lower trophic positions which are increasingly more herbivorous. Trophic position rarely provides information on specific dietary items[2,18] or individual variation in niche breadth[35] but rather quantifies the consumption of animal matter relative to other individuals in a population of omnivores. We analyzed annual trophic positions from $\delta^{15}N$ values in brown bear hair keratin[36]. Hair $\delta^{15}N$ represents a dietary integration of about a month (i.e., growing hair in June reflects the diet intake since May[37]). Bear hair is annually renewed through molting in June, regrows over the summer and fall, and stops growing during winter hibernation (Fig. 1A[38,39]). Guard hair samples collected in spring therefore reflect annual estimates of the cumulative protein intake of individuals during the previous active foraging season[38].

Using repeated samples of known mother-daughter pairs, we first estimated the extent of dietary specialization as permanent between-individual variation and second fitted a spatially explicit Bayesian hierarchical model (i.e., ´animal model´[7,40,41]) to quantify its sources. Specifically, we accounted for environmental similarity, with pairwise habitat similarity in individual bear home ranges encompassing the proportion of mature habitat, disturbed habitat, and habitat diversity (Supplementary Fig. 1). We further accounted for genetic heritability with a pedigree, for maternal effects by incorporating the mother's ID as a random effect, and for social learning as the fixed effect of a mother's trophic position on her offspring's trophic position. We allowed the effect of social learning to shift with time since offspring gained independence to account for individual learning later in life. We determined maternal trophic positions from a population-wide model accounting for sexual dimorphism, age, and permanent between-individual variation in diet (Supplementary Note 2). We validated that a similarity between offspring and mother trophic position reflected social learning during rearing, and found that their trophic positions were highly correlated when together in the first year of life (Supplementary Note 3). As we were interested in lifelong variation of dietary niche, and male offspring were only monitored for a short period after family breakup, we primarily focused on individual specialization of female offspring. However, we provide an additional reduced analysis including the relationship between maternal and male offspring trophic position in the two four years after family breakup and of the relationship between paternal trophic position and offspring trophic position. We also provide an alternative analysis accounting for spatial

correlation via spatial distance between home ranges instead of environmental similarity (Supplementary Note 5). We further fitted reduced models excluding either the effect of environmental similarity or social learning, respectively, to test whether spatial and genetic effects (Supplementary Note 6) or social and genetic effects (Supplementary Note 7) were confounded in philopatric female bears. Last, we validated our effect of social learning by refitting the model to a reduced dataset with observed maternal trophic positions during rearing, instead of modeled-averaged maternal trophic positions (Supplementary Note 8).

## Results

### Individual dietary specialization

We analyzed annual trophic positions from 213 hair samples collected from 71 female brown bears born to 33 unique mothers (median 2 daughters; range 1–6 daughters per mother). Repeated sampling (median 3 years; range 1–11 years) showed that female trophic position was unaffected by age (median [mean, 89% equal tails credible interval] explained variance = 1% [1%, 0–4%]; Supplementary Note 1) and

that individuals showed long-term individual specialization, accounting for 48% [47%, 31–63%] of the total variance in trophic position (Fig. 2). Individual variability in trophic position spanned half a trophic position ranging from 2.7 to 3.1 for individual females (Fig. 1B), which is equivalent to the difference between an omnivore feeding on a mix of plants and animal prey and a carnivore feeding predominantly on animal prey.

### Drivers of dietary specialization in female brown bears

Individual specialization was primarily driven by social learning, environmental similarity, and maternal effects (Fig. 2, Supplementary Table 1). Maternal trophic position dynamic over time since separation accounted for 13% [13%, 5–23%] of the total phenotypic variation in trophic position, while environmental similarity accounted for 5% [13%, 0.1–48%]. Additionally, maternal effects accounted for 11% [13%, 0.5%–31%] of variation in trophic position, indicating that siblings (full and half) of the same mother were more similar in trophic position throughout life compared to non-siblings. A remaining 9% [11%, 0.2–28%] of variance in trophic position was attributed to permanent

**A**

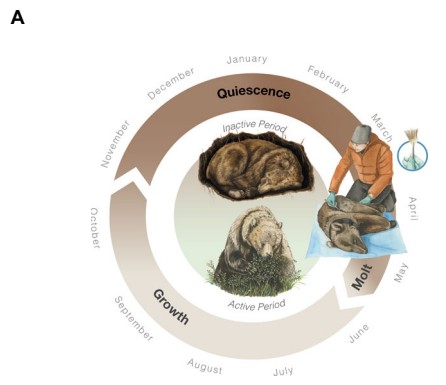

**B**

Posterior daughter trophic position

**Fig. 1 | Hair molt cycle and individual specialization. A** Bear hair generally grows from June until October and stable-nitrogen isotopes (δ15N) reflect cumulative diet intake during the period of hair growth. The quiescent phase, when hair ceases growing, lasts through hibernation, followed by emergence from the winter den and molting in late May-early June. Hair samples were taken during bear captures in

April–June and reflect the bears' diet in the previous year; **B** Posterior distribution of female trophic niche (bold line) and individual dietary niches indicated by each individual's posterior trophic position (modeled distribution with individual posterior medians indicated by black dots). Scientific illustration by Juliana D. Spahr, SciVisuals.com.

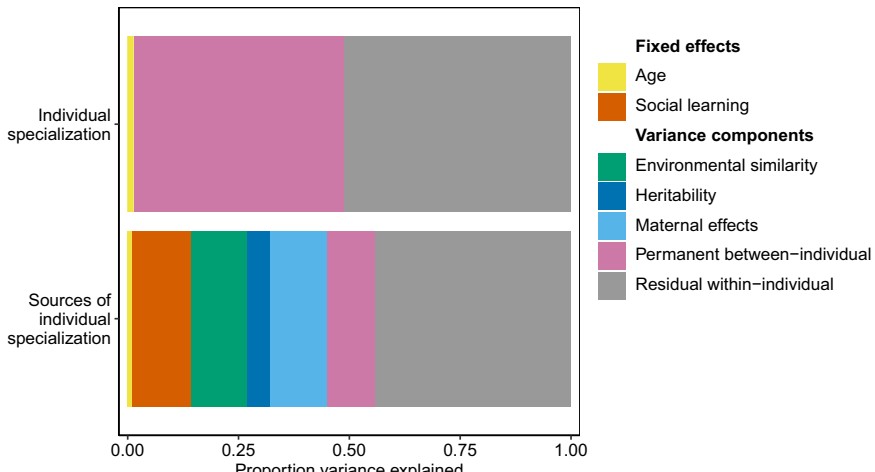

**Fig. 2 | Sources of individual specialization.** Individual specialization accounted for 48% of the phenotypic variation in the trophic position of female brown bears in Central Sweden. Trophic position did not change with age. We determined the proportion of variance (mean of the posterior distribution) explained by different

sources of individual specialization: Offspring social learning from the mother, environmental similarity, genetic heritability, maternal effects, permanent between-individual effects, and residual within-individual components.

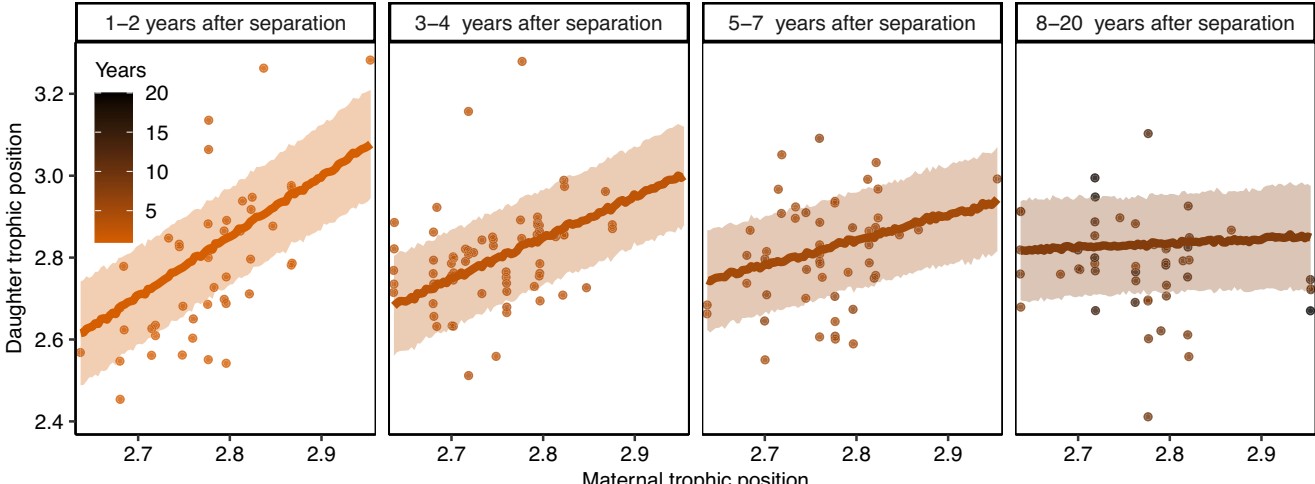

**Fig. 3 | Social learning.** Relationship between female brown bear trophic position and their mother's trophic position over the number of years separation from the mother, which occurs at 1.5–2.5 years of age in our population. The daughter's trophic position resembled their mothers' trophic position in the first years after separation but this similarity ceased after 4 years. Lines indicate predicted posterior mean estimates with ribbons corresponding to the estimated standard error, raw data are shown as points. Source data are provided as a Source data file.

between-individual effects (Fig. 2). Genetically more closely related individuals (including paternal half-siblings, aunts, or cousins) did not share a more similar trophic position (3% [5%, <0.1%–18%] of variance explained), providing no evidence that dietary specialization is heritable in this population (see also Supplementary Note 6, 7 assessing collinearity among variance components and Supplementary Note 9 assessing statistical power to detect additive genetic effects).

After family breakup, female offspring initially maintained a similar trophic position to their mother (Pearson's $r$ (42) = 0.66, $p < 0.01$ in the first two years after separation), which gradually became more dissimilar over time (Pearson's $r$ (66) = 0.31, $p = 0.01$ in year 3–4 after separation, Fig. 3). In the first years, offspring of more carnivorous mothers also had a higher trophic position while offspring of less carnivorous mothers had a lower trophic position. About five years after the separation from the mother, this correlation ceased to exist (Fig. 3). Additionally, daughters of the same mother (i.e., full- and maternal half-siblings) occupied similar dietary niches with consistently lower or higher trophic positions (Fig. 4). Bears inhabiting home ranges with a similar composition of mature and disturbed forest, as well as a similar habitat diversity in the home range, also had more similar trophic positions. The distance between pairwise home range centroids ($n = 5112$) ranged from 0.63 to 170 km with a median pairwise distance of 49 km and individuals living in closer proximity had a more similar trophic position than individuals living farther apart (median explained variance = 63%, Supplementary Note 5). However, after excluding spatial distance, social learning, and maternal effects, but not heritability, explained more variance in trophic position (Supplementary Note 6), indicating that spatial proximity and maternal effects are strongly related in this female philopatric species, where settlement home ranges of daughters are often close in space to their mothers forming so-called matrilineal assemblages. Social learning, i.e., the effect of maternal trophic position on offspring trophic position, was not confounded with additive genetic effects (Supplementary Note 7).

### Sex-specific social learning and paternal effects

Using trophic positions of both male and female offspring in the first two years after separation ($n_{Sons} = 37$, $n_{Daughters} = 49$) we found no evidence that the effect of social learning on offspring trophic position was sex-specific. In a mixed model, evaluating the effects of maternal trophic position, sex of the offspring (son or daughter), and their

interaction, leave-one-out-cross-validation (loo) indicated that neither the interaction nor the main term of sex improved the model (both elpd differences <4). Maternal trophic position as the sole predictor was the most parsimonious model and it explained 27% of the variance in offspring trophic position in the first two years of independence (Fig. 5A, Pearson's $r$ (84) = 0.51, $p < 0.001$), corroborating that social learning from the mother during rearing determines foraging behavior in the early years after family breakup in a similar fashion for male and female offspring. Further, using the posterior trophic position of the father as a covariate ($n = 40$ offspring sired by 17 unique fathers), we did not find that offspring trophic position was affected by paternal trophic position (Fig. 5B, explained variance = 1%, Pearson's $r$ (38) = 0.12, $p = 0.45$). While the modeled maternal trophic position correlated strongly with her observed trophic position in any given year (Supplementary Note 2), social learning explained even more of the phenotypic variance in daughter's trophic position (22% [7%–37%] instead of 13%) when fitting the observed maternal trophic position during rearing, instead of the modeled posterior average maternal trophic position to a reduced dataset (62 hair samples collected from 38 daughters, Supplementary Note 8). Our estimates of offspring social learning from the mother are therefore likely conservative and may underestimate the true effect of social learning on individual specialization.

### Discussion

Our multigenerational dataset reveals unique insights into the ontogeny of individual dietary specialization along a continuum from a more herbivorous to a more carnivorous diet in a long-lived omnivore. Specifically, the foraging strategy of offspring was intimately tied to the foraging strategy of their mother, a relationship that lasted up to four years after independence. We interpret this relationship as evidence that social learning plays an important role in shaping an individual's dietary specialization. Five years into independence, the similarity between the trophic position of mothers and daughters slowly faded, likely due to individual learning and experience during solitary life. In addition, offspring of the same mother also shared similarities in their trophic position, potentially mediated through maternal genetic or environmental effects on body size[31]. Additive genetic effects on the other hand were not significant, providing no evidence for the heritability of dietary specialization in this population. Similar to the effect of social learning from the mother fading over

time, additive genetic and maternal effects could be life-stage specific with maternal effects being more influential in juveniles[13], although evidence for this is mixed[10]. We were not able to quantify life-stage-specific heritability and maternal effects due to sample size limitations. In general, previous ecological studies have mainly concentrated on resource availability as the main driver of resource selection[42] and individual specialization[4]. However, our results show that, within populations, the environment is only one of several components shaping individual variation in dietary niches. We conclude that social learning of maternal dietary preferences during early-life ontogeny and maternal effects (i.e., maternal genotype and environment), which together explained about 24% of the variation in trophic position, play a pivotal role in spreading and maintaining feeding strategies within populations, even in species with otherwise solitary lifestyles. In addition, variation linked to permanent between-individual effects (in our study 9%) could be associated with either uncontrolled variation in resource availability in the environment (i.e., ecological opportunity[4,35]) or individual differences in resource preference. The latter could, for example, be caused by individual learning

later in life and demonstrates the potential for behavioral innovation in this population. Ultimately, between-individual variation in dietary specialization allows populations to adapt to changes in resource availability, such as new invasive prey or declines in food items due to climate change.

Our findings are particularly relevant for species in which dietary specialization impacts individual fitness[20,35,43]. For example, protein-rich diets may promote greater offspring survival or mass gain[44]. Social learning in general, therefore, presents an important, yet under-studied, pathway by which alternative behavioral strategies can establish and spread more rapidly within populations than by genetic evolution alone[45]. Species more adept in social learning of dietary strategies may therefore show greater behavioral variability at the population level, which could give them an advantage when adapting to changing environments due to landscape modification or urbanization, climatic variations, or global change in general. Moreover, there is evidence that the strength of social learning in shaping individual phenotypes is not only species-specific but can also vary among populations or individuals of the same species[12,46].

Our research also points to several aspects of social learning that warrant future research. First, there is little information on whether maternal care and social learning tend to be more prevalent in species or populations with greater dietary specialization. There is some evidence that within populations, dietary generalists (i.e., those with a wider dietary niche) seem to provide more intense parental care[47] than their conspecific dietary specialists (i.e., ones with a narrower dietary niche), but the link to social learning of foraging preferences remains unclear. Second, while generalist species with a wide ecological niche have been frequently shown to be more successful under changing environmental conditions, such as urban environments or fragmented landscapes, than specialist species[48–50], it is currently unknown whether this success could be partially mediated by social learning. Finally, social learning could alternatively limit behavioral innovation and adaptation due to adherence to social traditions[51]. We therefore suggest that alternative hypotheses should be evaluated that consider how social learning impacts individual specialization and in turn the adaptability of species under global change.

Our findings that dietary specialization can be socially learned and transmitted are particularly relevant for species where individual specialization is related to human-wildlife conflict[52]. For example, the removal of single individuals who are known to cause conflict is an effective strategy to halt the spread of problematic behavior and mitigate the conflict, while minimizing the impact on species

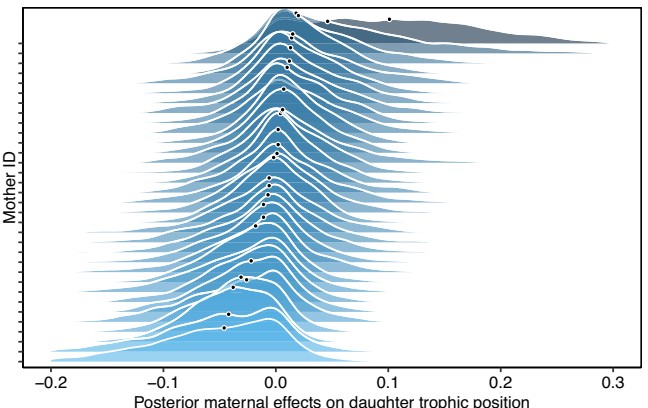

**Fig. 4 | Maternal effects.** Additional maternal effects (e.g., maternal genotype or maternal environment) explained further similarities in trophic position among daughters of the same mother (and differences between daughters of different mothers). Densities correspond to the mother's posterior trophic position with each mother's posterior medians indicated by black dots. Shadings from light to dark correspond to mothers producing daughters with lower to higher trophic positions.

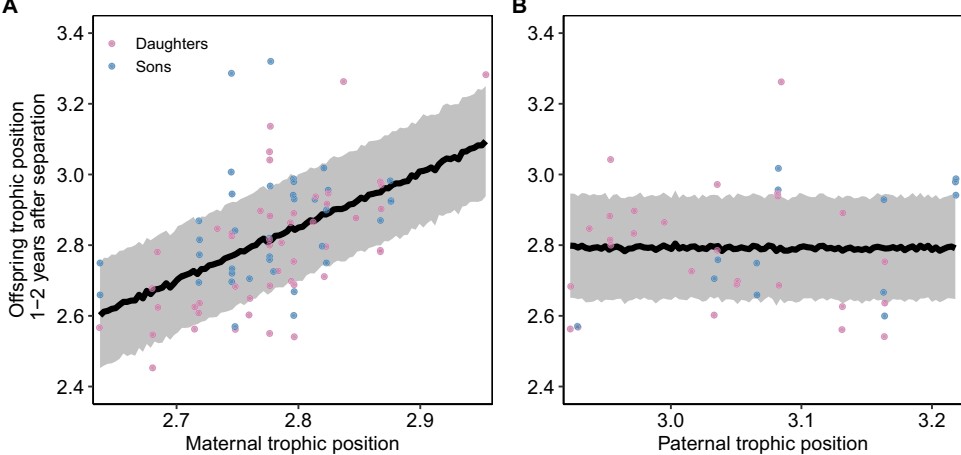

**Fig. 5 | Sex-specific social learning & paternal effects.** **A** Trophic position of male ($n = 37$) and female ($n = 49$) offspring in the first two years after separation was equally correlated to maternal trophic position suggesting that social learning is not sex-specific. **B** Offspring trophic position in the first two years after separation was not correlated with the trophic position of their father ($n = 17$ unique fathers). Source data are provided as a Source Data file.

conservation goals[52]. Foraging behavior that causes conflict with humans has also been shown to change in ursids over their lifetimes, remarking the crucial role of individual plasticity in behavior[53]. Social learning of behavior from the mother[54], including individual specialization and foraging on anthropogenic food resources, has been previously observed in ursids[55–58]. However, none of these studies tracked offspring's diet over their lifetimes or were able to simultaneously account for the mother's diet, genetics, environment, and other maternal effects that could explain similar patterns of individual specialization. While some of the aforementioned studies suggest either the environment or social learning as primary drivers of individual specialization, we suggest using caution in assigning causality in dietary specialization, when potentially confounding alternative sources cannot be accounted for. Specifically, in female-biased philopatric species, spatial proximity does not only encode for spatial variation in resource abundance but is also conflated with closely related individuals sharing space. In brown bears, some daughters settle close to their mother's home range[34] creating spatial clusters of closely related females, so-called matrilinear assemblages[59]. Due to the spatial dependence of these assemblages, it can therefore be difficult to untangle social learning from the mother from other maternal effects (i.e., maternal genotype or maternal environment), or the ambient environment. Our study population spanned over 170 km with spatial proximity explaining 63% of the total phenotypic variation in the trophic position of female bears: individuals further apart tended to have more different diets. However, when replacing spatial proximity with environmental similarity among home ranges, the explanatory power was attributed to social learning and maternal effects along with the environment, while when omitting the social learning effect, power was attributed to the environment. Our results therefore demonstrate that individual dietary specialization is not caused by a single driver in isolation but the product of many factors, namely social learning, maternal effects, and the environment.

Our finding that social learning has a similar or stronger impact on resource selection as the environment provides important insights for a range of studies on habitat selection, dispersal, and range expansion. For example, a popular theory known as "natal habitat preference induction" suggests that dispersing animals select areas for settlement that resemble their natal habitat, even at fine habitat scales[30]. Our results challenge the notion that habitat similarity alone drives natal settlement strategies and rather suggest that socially learned diet preferences, and hence the selection for food resources themselves, could play an important role in producing similar patterns of settlement selection like induced natal habitat preferences. Recent studies of migration and short stopover behavior in whooping cranes (*Grus americana*) have also observed that social learning rather than environmental conditions[60] or genetic heritability[61] led to the emergence and establishment of alternative migratory behavior. Similar to what our study shows with respect to individual specialization, social learning of migration strategies primarily determined behavior in early life whereas individual-experiential learning shaped behavior later in life[62].

Drivers of individual dietary specialization are well documented among populations of the same species. However, systematic studies delineating the sources of individual specialization within populations are lacking, likely because suitable datasets including multigenerational, genetic, environmental, and life-history information are rare. We show that, in addition to the environment, social learning and maternal effects can be important sources of dietary specialization.

## Methods
### Permits
All animal captures and handling were performed in accordance with relevant guidelines and regulations and were approved by the Swedish authorities and ethical committee (Uppsala Djurförsöksetiska Nämnd: C40/3, C212/9, C47/9, C210/10, C7/12, C268/12, C18/ 15. Statens Veterinärmediciniska Anstalt, Jordbruksverket, Naturvårdsverket: Dnr 35-846/03, Dnr 412-7093-08 NV, Dnr 412-7327-09 Nv, Dnr 31-11102/12, NV-01758-14). Samples were collected and stored in Sweden and shipped to Poland and Canada for preparatory procedures and stable isotope analyses. CITES permits were obtained to ship samples (permitting numbers: 16PL000376/WP and 15PL000102/WP).

### Bear hair sample collection
We collected brown bear hair samples in south-central Sweden (~N61°, E15°) as part of a long-term, individual-based monitoring project (Scandinavian Brown Bear Research Project; www.bearproject.info). Hair samples were collected from known individuals and their offspring during captures in spring (April–June) 1993–2016. Bears were immobilized by a helicopter and were fitted with a VHF or GPS collar. A vestigial premolar tooth was collected from all bears not captured as a yearling to estimate age based on the cementum annuli in the root[63]. Tissue samples (stored in 95% alcohol) were taken for DNA extraction to assign parentage and construct a genetic pedigree[59]. Guard hairs with follicles were plucked with pliers from a standardized spot between the shoulder blades and archived at the Swedish National Veterinary Institute. Bear cubs are born in January or February during winter hibernation and are typically first captured together with their mother as yearlings at the age of ~15 months. Cubs in this population separate from their mother during the mating season in May or June after 1.5 or 2.5 years[64]. A hair sample taken in spring reflects the summer-fall diet of the bear in the previous active season (Supplementary Fig. 2). Only hair samples of solitary, independent offspring taken in spring at least 10 months after separation from the mother were included in this study.

### Sample selection
As we were interested in the drivers of lifelong variation of dietary niche, and male offspring were only monitored for a short period after family breakup, we focused our main analysis on repeated sampling of female offspring after separation from their mother. We only included female offspring with complete information of predictor variables: known age, known maternal identity represented by at least one stable isotope sample, information about home range location after independence, and occurrence in the study population genetic pedigree ($n_{ID} = 71$, $n_{Samples} = 213$). We additionally fitted two reduced analyses delineating (a) whether the effect of social learning was sex-specific by including male offspring trophic position in the first 2 years after separation ($n_{male} = 37$, $n_{male} = 49$), and (b) testing for a relationship between paternal trophic position and offspring trophic position ($n = 40$ from 17 unique fathers). In the supplementary information, we provide an analysis including all samples of independent males ($n = 98$, $n_{Samples} = 219$) and female bears ($n = 115$, $n_{Samples} = 335$, Supplementary Note 2). This model served to delineate posterior maternal and paternal trophic positions which were used as predictor variables in the main analyses. We also validated that maternal and daughter trophic positions were correlated (using Pearson's correlation coefficient) during rearing ($n = 116$ mother-daughter pairs), providing the basis for a social learning effect after independence (Supplementary Note 3).

### Moose sample collection
We collected samples of the natural foods most important for brown bears in the study area (Supplementary Fig. 2), including 21 samples of moose hair (*Alces alces*), the most common meat source in the diet of brown bears in our study area[65] in the spring-autumn field season of 2014. Samples were placed in a paper envelope and dried at ambient temperature.

## Stable isotope analyses

Both moose and bear hair samples were rinsed with a 2:1 mixture of chloroform:methanol or washed with pure methanol to remove surface oils[66]. Dried samples were ground with a ball grinder (Retsch model MM-301, Haan, Germany). We weighed 1 mg of ground hair into pre-combusted tin capsules and combusted them at 1030 °C in a Carlo Erba NA1500 elemental analyzer. $N_2$ and $CO_2$ were separated chromatographically and introduced to an Elementar Isoprime isotope ratio mass spectrometer (Langenselbold, Germany). Two reference materials were used to normalize the results to VPDB and AIR for $\delta^{13}C$ and $\delta^{15}N$ measurement, respectively: BWB III keratin ($\delta^{13}C = -20.18‰$, $\delta^{15}N = 14.31‰$, respectively) and PRC gel ($\delta^{13}C = -13.64‰$, $\delta^{15}N = 5.07‰$, respectively). Measurement precisions as determined from both reference and sample duplicate analyses were ±0.1‰ for both $\delta^{13}C$ and $\delta^{15}N$.

## Bear trophic position

We calculated the trophic position of each bear hair sample relative to the average $\delta^{15}N$ value of moose hair representing trophic level 2 (mean ± sd = 1.8 ± 1.26 ‰, $n = 21$, Supplementary Fig. 2). Bears consume most of a moose carcass, including meat, skin, and hair. Soft tissue samples of moose carcasses could not be obtained but according to the literature the ratio of $\delta^{15}N$ in ungulate hair to meat ranges between 0.77‰–1.0‰. (see S3.1 in ref. 67). We consider trophic positions calculated from moose hair representative and a correction of the $\delta^{15}N$ moose hair signature would only add an arithmetic correction but not change the distribution of bear trophic positions. The trophic position is calculated as (Eq. 1) the discrepancy of $\delta^{15}N$ in a secondary consumer and its food source divided by the enrichment of $\delta^{15}N$ per trophic level, plus lambda, the trophic position of the food source (e.g., 1 for primary producers, 2 for primary consumers, 3 for secondary consumer, 4 for tertiary consumers)[68]. We used an average trophic enrichment factor of 3.4‰[68] and added a lambda of 2 given the moose baseline trophic position as a strict herbivore.

$$\text{Bear trophic position} = (\delta^{15}N_{Ursus\ arctos} - \text{average}(\delta^{15}N_{Alces\ alces.hair}))/3.4 + 2 \quad (1)$$

Under an omnivorous diet including the consumption of herbivores (in particular moose but also ants such as *Formica* spp., *Camponotus herculeanus* with average $\delta^{15}N$ indistinguishable from moose, Fig. S1), bear trophic position values were expected to fall between 2 and 3. Values approaching 4 indicate a trophic enrichment through the consumption of other omnivorous or carnivorous animals. Because absolute trophic position values by definition depend on the $\delta^{15}N$ of the food source used to calculate them, the values reported in our study should not be used for comparing the degree of carnivory in our study to other study systems and populations.

## Sources of individual variation in trophic position

**Environmental similarity.** Resources may not be distributed evenly in space. For moose, population density and hunting quotas (which determine the availability of slaughter remains) vary across the study area. For ants, the availability of old forests and clearcuts determine their abundance[69]. Furthermore, brown bear daughters are often philopatric with limited dispersal and settle close to their mother's home range[34]. The median dispersal distance of daughters, namely the distance between natal and settlement home range centroids in this study was 8.56 km (range 1.4–28.8 km). Genetic, spatial, and social learning effects may therefore be confounded with related bears occupying adjacent ranges with similar resource availability. Elsewhere, accounting for environmental similarity through spatial autocorrelation in animal models has revealed that a major portion of variance may be attributed to environmental similarity rather than genetic heritability[7,40,70], but see also ref. 71. Here, we accounted for

environmental similarity by extracting habitat composition in each bear's lifetime home range ($n = 71$, Supplementary Fig. 1). We fitted individual movement models and constructed 95% home ranges using the autocorrelated kernel density estimator in the R package ctmm[72]. Bears were monitored for a minimum of 2500 GPS locations ($n = 47$) or were located via VHF on at least 25 days ($n = 24$). The median lifetime home range size was 241 km², which is comparable to a circle with a 17.5 km diameter. We used a Corine landcover map (25 m resolution) which we updated annually with polygons of newly emerged clearcuts (data obtained from the Swedish Forest Agency). We extracted home range habitat composition in the year when the diet was assessed. When individuals were monitored for multiple years, we extracted the home range composition for the median year. Annual changes in home range habitat composition were negligible (Supplementary Note 4). We calculated the proportion of mid-aged and old forests and the proportion of disturbed forests (clearcuts and regenerating young forests) within the 95% utilization distribution. Additionally, we calculated habitat diversity using the Simpson diversity index from the R package landscapemetrics[73]. Following Thomson et al. [40], we calculated the Euclidean distance between scaled and centered habitat composition and habitat diversity in multivariate space, assuming equal importance of each component. Pairwise distances were scaled between 0 and 1, where increasing values indicated more similar habitat composition. Spatial autocorrelation of home range habitat composition seized after 10–15 km, which is less than the diameter of a median home range (Supplementary Note 4). In the supplementary material, we provide an alternative analysis accounting for spatial autocorrelation of individual dietary niches with a pairwise spatial distance matrix of home range centroids (S matrix; Supplementary Note 5).

**Genetic pedigree.** A genetic pedigree based on 16 microsatellite loci was available for the population including 1614 individual genotypes, spanning six generations[59]. All female offspring and mothers in this study were genotyped and included in the population's genetic pedigree. We used Cervus 3.0[74] for the assignment of fathers and COLONY[75] for creating putative unknown mother or father genotypes and sibship reconstructions (see for details in ref. 59).

**Maternal identity.** The study was based on a population of marked females and their offspring. Therefore, all mothers included in this study were known from observations of mother-offspring associations.

**Maternal (and paternal) trophic position.** Based on repeated hair samples of 115 female ($n_{female} = 335$) and 98 male ($n_{male} = 219$) bears, we fitted a linear mixed-effects model for female and male bears respectively, to estimate sex-specific between-individual variation in trophic position (Supplementary Note 2). We modeled trophic position as a function of a quadratic relationship with age and we controlled for individual random intercepts. We concentrate on the relationship between age and trophic position as, unlike mass or size[31], age is not confounded with between-individual effects (i.e., age itself cannot be heritable unlike mass or size). However, we also show the relationship between mass and trophic position in Supplementary Note 1. We estimated repeatability, i.e., variance standardized individual variation, as among-individual variance divided by total phenotypic variance. We extracted the variance in fitted values (variance explained by fixed effects), among-individual, and residual variance and calculated Nakagawa's marginal and conditional R2. The female trophic position did not vary with age but was highly repeatable over multiple years. Age accounted for 26% of the variation in male trophic position and male trophic position was moderately repeatable. For all daughters, we extracted their mother's (and father's) trophic position as the median of the posterior distribution of their respective random intercept. The

modeled maternal posterior trophic position and the observed maternal trophic position in a given sampling year were strongly positively correlated (Pearson correlation coefficient $r = 0.78$, $t = 22.63$, df = 336, $p < 0.001$, Supplementary Note 2).

## Statistical analysis

**Main analysis: sources of dietary specialization in brown bear daughters.** We applied a two-step modeling approach to our final dataset of 71 female offspring with 213 repeated annual measures of trophic position. First, we fitted a *basic* linear mixed-effects model to estimate individual dietary specialization as permanent between-individual variation ($V_I$) in trophic positions. For this, we used repeated measures of the same individual and fitted individual random intercepts. We accounted for a nonlinear effect of age (second-order polynomial, scaled by the standard deviation). We extracted the variance in fitted values ($V_{Age}$; variance explained by age), permanent between-individual ($V_I$), and residual within-individual variance ($V_R$) and estimated each component's proportional contribution to the total phenotypic variance ($V_P = V_{Age} + V_I + V_R$) through variance standardization[76].

Second, we used a spatially explicit Bayesian hierarchical model (i.e., 'animal model')[5,40,41] to partition permanent between-individual variance ($V_I$) in trophic position into its sources; the fixed effects age ($V_{Age}$) and social learning ($V_{SL}$), and the variance components permanent between-individual variance ($V_I$), environmental similarity ($V_E$), additive genetic variance ($V_A$), maternal effects ($V_M$), and residual within-individual variance ($V_R$). We followed the "hybrid" strategy suggested by McAdam, Garant[14] and tested for social learning of trophic position from the mother ("social learning") by incorporating maternal trophic position as a fixed effect into the model. Thus, $V_M$ pools the remaining phenotypic variation of offspring trophic position that cannot be explained by maternal trophic position. Since age and time since separation were perfectly correlated (Pearson correlation coefficient >0.99) we accounted for age with a nonlinear effect of time since separation between mother and daughter (second-order polynomial, scaled by the standard deviation). We further accounted for a decrease in the social learning effect over time by fitting an interaction between maternal trophic position and time since separation.

We calculated the total variance explained by the model ($V_P = V_{Age} + V_{SL} + V_I + V_E + V_A + V_M + V_R$) and calculated the proportion of the total variance explained by each model component. For the fixed effects, we partitioned the variance explained into $V_{SL}$ (i.e., maternal trophic position and its interaction with time since separation) and $V_{Age}$ (i.e., the main effect of time since separation), respectively, by calculating the independent contribution of each component to the total variance explained by the fixed effects, following the approach by Stoffel, Nakagawa[77] adapted to a Bayesian framework (see code under[78]). For all parameters, we report the median and mean as measures of centrality and 89% credible intervals, calculated as equal tail intervals, as measures of uncertainty[79,80]. We deemed explained variance proportions as inconclusive when the lower credible interval limit was <0.001 (i.e., <0.1%)[81].

In Supplementary Note 5 we fitted an alternative model in which we substituted the environmental similarity matrix with a spatial distance (S matrix)[7]. We extracted centroids from lifetime home ranges and calculated a pairwise Euclidian distance matrix between all bear home range centroids to account for spatial autocorrelation driven by spatial proximity of home ranges. We then refit our main model including spatial distance ($V_S$) instead of environmental similarity ($V_E$). We further fitted a set of reduced models to assess collinearity between genetic and permanent maternal effects with spatial proximity (Supplementary Note 6) and collinearity of social learning and additive genetic effects (Supplementary Note 7). For this we compared the full model to models leaving out (a) any effect of spatial distance or

environmental similarity, (b) spatial effects and permanent maternal effects (Supplementary Note 6), and (c) leaving out maternal trophic position. Last, we performed a power analysis to assess whether our dataset was large enough to detect significant additive genetic variance (Supplementary Note 9). We used a permutation approach recently suggested by Pick, Kasper[82] to generate a $p$-value for additive genetic variance. We fitted a reduced "basic animal model", controlling only for bear ID and genetic structure, i.e., omitting social learning, environmental similarity, and maternal effects. Keeping the architecture of our pedigree (Dam/Sire pairs) but randomly assigning parents to offspring, we permutated our dataset 1000 times, fitting 1000 animal models as null distribution. If related individuals have similar dietary specialization (i.e., heritability of dietary specialization), the observed pedigree should explain more variance than the permuted pedigrees. We thus generated a $p$-value by calculating the proportion of permuted models where the explained variance was larger than the explained variance in the observed dataset.

**Sex-specific effect of social maternal learning.** To evaluate whether the effects of social learning on offspring trophic position are sex-specific, we fitted a set of reduced mixed-effects models to male and female offspring trophic position estimates from the first two years after separation: we fitted a model controlling for maternal trophic position and interaction with the sex of the offspring (male/female), an additive effect of sex of the offspring, or no effect of offspring sex. We controlled for repeated measures from the same individual with a random intercept for bear id. We compared models using leave-one-out-cross-validation (loo) to determine the most parsimonious model. We also determined the variance explained by the fixed effect of maternal trophic position (using the R package "performance"[83]) and computed the Pearson correlation coefficient between the trophic position of offspring and mothers.

**Paternal effects of dietary specialization.** To evaluate paternal effects on offspring trophic position we fitted a mixed-effects model with offspring trophic position in the first two years after separation as response and the posterior trophic position of the father (Supplementary Note 2) as predictor, while controlling for repeated measures with a random intercept for bear id. We compared the model to a null model using leave-one-out-cross-validation (loo), determined the variance explained by the paternal trophic position, and computed the Pearson correlation coefficient between the paternal and offspring trophic position.

All models were fitted with a Gaussian family using the R package "brms"[84] based on the Bayesian software Stan[85,86]. We ran four chains to evaluate convergence which were run for 6000 iterations, with a warmup of 3000 iterations and a thinning interval of 10. All estimated model coefficients and credible intervals were therefore based on 1200 posterior samples and had satisfactory convergence diagnostics with $\hat{R} < 1.01$, and effective sample sizes >400[87]. Posterior predictive checks recreated the underlying Gaussian distribution of trophic position well. All statistical analyses were performed in R 4.4.1[88].

## Reporting summary

Further information on research design is available in the Nature Portfolio Reporting Summary linked to this article.

# Data availability

The primary data generated in this study have been provided in the OSF repository under accession code https://doi.org/10.17605/OSF.IO/68B9U[78]. The raw GPS & VHF location data are available under restricted access for sensitivity reasons, access can be obtained from J.K. through correspondence with the first author (A.G.H.). Source data are provided with this paper.

## Code availability

Code to reproduce all analyses are provided in the OSF repository; under the accession code https://doi.org/10.17605/OSF.IO/68B9U[78].

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

## Acknowledgements

A.G.H. has received funding from the European Union's Horizon 2020 Research and Innovation Programme under the Marie Skłodowska-Curie Grant agreement No 793077 and from the German Science Foundation (HE 8857/1-1). The study was further funded by the Norway Grants under the Polish-Norwegian Research Programme administered by the National Research Centre for Research and Development in Poland and the Norwegian Research Council (J.A., N.S., A.S., and A.Z.; GLOBE No POL-NOR/198352/85/2013). Isotope analyses were funded through a Robert Bosch Foundation grant to TM and the GLOBE project and conducted by KAH (with assistance from Blanca Xiomara Mora Alvarez and Geoff Koehler), D.J. and A.S. We thank the Scandinavian Brown Bear Research Project (SBBRP) for providing access to the data. The SBBRP was funded by the Norwegian Environment Agency, the Swedish Environmental Protection Agency, the Austrian Science Fund, and the Norwegian Research Council.

## Author contributions

A.G.H., J.A., and T.M. developed the work. A.Z., J.K., K.A.H., N.S., and A.S. provided the data. A.Z. managed the sample collection. A.S. managed the hair samples database and prepared samples for stable isotope analyses by K.A.H. D.M.J. provided laboratory space and resources and supervised preparatory procedures. S.C.F. constructed and provided the genetic pedigree. J.E.H. calculated home ranges and centroids. A.M. and J.A. advised to the analysis and interpretation of stable isotope data. T.M., N.S., and A.Z. secure project funding. A.G.H. performed the statistical analyses with input from J.A. A.G.H. wrote the manuscript with help from T.M., J.A., and input from all authors.

## Funding

## Competing interests

The authors declare no competing interests.

## Additional information

[1]Behavioural Ecology, Department of Biology, Ludwig-Maximilians-Universität in Munich, Planegg-Martinsried, Germany. [2]Senckenberg Biodiversity and Climate Research Centre (SBiK-F), Frankfurt (Main), Germany. [3]Institute of Nature Conservation, Polish Academy of Sciences, Krakow, Poland. [4]Departamento de Ciencias Integradas, Facultad de Ciencias Experimentales, Centro de Estudios Avanzados en Física, Matemáticas y Computación, Universidad de Huelva, Huelva, Spain. [5]Estación Biológica de Doñana, Consejo Superior de Investigaciones Científicas, Sevilla, Spain. [6]Environment and Climate Change Canada, Science and Technology, Saskatoon, SK, Canada. [7]Department of Biology and Advanced Facility for Avian Research (AFAR), University of Western Ontario, London, ON, Canada. [8]Department of Veterinary Biomedical Sciences, Western College of Veterinary Medicine, University of Saskatchewan, Saskatoon, SK, Canada. [9]Institute of Geosciences, Goethe University Frankfurt, Frankfurt (Main), Germany. [10]Norwegian Institute for Nature Research, Trondheim, Norway. [11]Department of Wildlife, Fish, and Environmental Studies, Swedish University of Agricultural Sciences, Umeå, Sweden. [12]Department of Natural Sciences and Environmental Health, University of South-Eastern Norway, Bø, Norway. [13]Institute of Wildlife Biology and Game Management, University of Natural Resources and Life Sciences, Vienna, Austria. [14]Department of Biological Sciences, Goethe University Frankfurt, Frankfurt (Main), Germany. [15]Deceased: Keith A. Hobson. ✉e-mail: hertel@biologie.uni-muenchen.de

