## [Peer Review file · Nature Communications]

Ontogeny shapes individual dietary specialization in female European brown bears (*Ursus arctos*)

Corresponding Author: Dr Anne Hertel

Version 0:

Reviewer comments:

Reviewer #1

(Remarks to the Author)

The paper was well written, and the results were interesting. The value of having this larger dataset was quite clear. I did have some queries that are mostly seeking clarity but suggest that these will improve the manuscript further.

Abstract:

L37: The apostrophe after “species” is not really necessary.

L39: Add a comma after “drivers”. Use the plural of remain (remains).

L42: After reading the manuscript, I’m not entirely clear on what you mean by “environmental forcings”. As you only use this term in the abstract (here and L50), I suggest using more relevant terminology that matches the main text.

L46-47: Rephrase: “the trophic position of offspring closely”.

L49: Replace “reveals” with “shows”.

Introduction:

My primary concern with the introduction is how maternal learning is differentiated from maternal effects, offspring learning, environmental effects and genetic effects. It seems to be blended together, and the definitions do not provide much clarity. I suggest using “offspring learning” rather than “maternal learning” if this is what you really mean (as suggested on L114). But this needs to be clearly differentiated from maternal dietary effects (e.g. offspring gaining information about food resources during suckling).

L61: Replace “hence” with “likely”.

L62: Add a comma after “composition”.

L63: Rephrase: “key for driving species resilience in response to”

L64: A comma may be useful after “world”.

L66 onwards: This paragraph was very long and incorporated many different ideas. The concept of environmental effects wasn't well developed in contrast to the other ideas. The issue with separating out maternal learning and maternal effects is that mothers can still affect offspring postnatally because of the environment they might be experiencing independently of their offspring. I suggest breaking this paragraph up and systematically discussing each topic clearly and unambiguously.

L66: The comma after “opportunity” is not necessary.

L68: Replace “yet” with “however”.

L69: Bring “been” after “rarely”.

L71: This should simply be parental effects. Fathers can also affect offspring development and behaviour even in species without paternal care. I suggest introducing the concept broadly, and then discussing maternal effects specifically within that context.

L74: Rephrase: “between the dietary phenotypes of mothers and their offspring”.

L75: Replace “are” with “may be”.

L78-79: Again, paternal effects may also be important. As suggested, restructuring this paragraph and then focusing on parental effects specifically will allow you to introduce the concept broadly and then refine it to explain why maternal effects are a potentially stronger factor for solitary species.

L81: Please provide a reference for this statement. It is not clear why maternal effects would be evident from the similarity of

repeated samples of siblings. Couldn't a genetic effect equally result in a similar response between siblings?
L82-83: Likewise, please provide a reference for this statement. It is not clear why similarity between mothers and offspring reflects maternal learning (which should be defined here because it's not clear) when similarity could be a consequence of genetic effects, maternal effects or offspring learning from their mothers.
L94: Add a comma before "we then attributed".
L99: I suggest "newborn ungulates" rather than "ungulates neonates".
L100: Hyphenate "plant-based".
L101: Delete the comma before "however" and replace with a semi-colon.
L106: Use plural "sources".
L107: I suggest starting a new paragraph before "Brown bears".
L110: Add a comma after "mother". Replace "assume" with "predict".
L112: Add "may" after "hence". Use plural "behaviors".
L113: Delete "hence". Replace "of" with "from".
L114: Delete the comma before "however" and replace with a semi-colon.
L116: But body size is also strongly influenced by environmental effects, such as resource availability, so could equally affect maternal effects through effects on maternal environment.
L117: Add a comma after "population".
L121: With reference to "leading to greater similarity" - This assumes no competition between siblings (either in utero or ex utero). Runts could be argued to gain fewer maternal resources, so why would you expect similar trophic positions if there are differences?
L123: I suggest starting a new paragraph before "To assess".
L128: Delete the comma after "may".
L132: Past tense: "fitted".
L133: Consider "untangle" rather than "disentangle".
L134: Delete "as determinants of" and replace with "to".
L136: Add a comma after "ranges".
L137 and elsewhere: I suggest putting all the "such as" parts in brackets as examples. "Mature habitat (e.g. old and mid-successional forests)" – and so on.
L142: Should this read: "a mother's trophic position on her daughters' trophic positions"?
L144: Why does 3 come first? Consider reordering to have the material presented in numerical order.
L146: Replace "last" with "Finally".
L150: Why is this in supplementary material only and not in the main text? Perhaps indicate here why you provide this information as supplementary, considering you make the point that you are focusing on daughters.
L156: Delete "last".

Results:

These were mostly easy to follow.

L162 Replace "in" with "from".
L164: Replace "revealed" with "showed".
L177: Delete "as".
L179: But in L176 you say that siblings were more similar in trophic position. So how do you know this is a maternal effect and not a genetic effect? Please elaborate.
L183: Replace "as" with "to".
L193: Start the point after the colon with a small letter "a".
L194: Commas before and after "but not heritability" could increase the clarity and flow of this sentence.
L195-196: Perhaps I missed something here, but I don't understand how controlling for spatial proximity showed that it was a confounding effect. Perhaps consider additional elaboration for clarity here.
L201: Rephrase: "offspring trophic position".

Discussion:

L246: Rephrase: "between the trophic position of mothers and daughters slowly".
L251: Delete the comma before "however" and replace with a semi-colon, adding a comma after "however".
L253: Using the term "early-life imitation" here is confusing. As suggested earlier, I suggest using a single term, offspring learning, in place of others (including maternal learning) so that the reader isn't left confused in the discussion.
L258: Replace "adapt" with "respond flexibly" as this is likely a better reflection of what is occurring, rather than adapting (in the classic evolutionary sense).
L275: Delete the comma after "parental care".
L278: Singular "remain".
L282: Delete "last" and replace with "finally".
L293: Rephrase: "over their lifetimes, further indicating the crucial".
L295: Add a comma after "resources".
L297: Delete the comma after "effects".
L300: Delete the comma after "specialization".
L306: Consider "untangle" rather than "disentangle".

Methods:

L369: Rephrase: "diet of brown bears in".
L379: Please indicate what times seed preference tests were conducted.
L391: Delete "that".
L423: Consider "Furthermore" rather than "further".
L461: Delete the comma after "deviation" and replace with a semi-colon.
L473: Past tense "fitted".

Supplementary material:

It was disappointing that most of the supplementary material was not referred to in the main text to provide context. I suggest adding the citations to the supplementary material at relevant and appropriate points in the text so that the reader has context and can expand their understanding as necessary.

Reviewer #2

(Remarks to the Author)

The authors investigated what mechanisms drive among-individual variation in diet composition (also known as individual specialization) in the European brown bear. To address this question, the authors used a complete dataset in combination with an appropriate and complex quantitative genetic approach. They showed that genetic heritability is weak and that most variations are explained by maternal effects (including maternal learning). To the best of my knowledge, this is the first study to estimate the heritability of dietary composition and to consider the many different confounding mechanisms (habitat similarity, maternal effect, permanent environment) at the same time. These results are very promising because they involve a species that can come into conflict with human populations due to the type of dietary specialization of certain individuals.

While I am convinced that this paper will move the field forward, I think it should benefit from some clarifications in the introduction and methods to help the reader through the paper. In addition, the authors have made an effort to account for a large number of confounding effects, but I think a first step would be to include a statistical power analysis (in the Supplement) to understand what the chances are of detecting a true heritability with a basic animal model. I have made other major and minor comments which I have grouped by sections.

Introduction:

I feel that the second paragraph needs to be rearranged to be clearer and more effective. It might not be easy to understand how the different mechanisms differ from each other. For example, it says maternal learning and maternal effect, but in my opinion maternal learning is part of maternal effect (and it is more or less what you say in the supplementary "maternal effects (both maternal learning and maternal effects)"). Regarding maternal learning mechanisms, I am confused by the use of interchangeable words between maternal imitation and learning. Imitation is a special case of learning, but it would be better to stick to one term. Regarding genetic inheritance mechanisms, I think that maternal genetic effect is a kind of (indirect) genetic inheritance.

I don't know what the recommendation is for the "brief material and methods" at the end of the introduction, but I think it's too detailed and I get lost between when you give your aims and results. You give the detailed methods at the end of the manuscript, so I won't spend too much time on it.

L58 add (a.k.a. individual specialization) or L60 add individual specialization (i.e. among-individual variation of niche).

L63 Are you talking about among or within individual variation?

L63-65 I think the flow in the first paragraph will be better if this sentence is moved right after the first one (L59).

L66-68 Does this apply to all types of individual specialization or just individual dietary specialization?

L-68- among-individual variation or both among and within?

L-70 Four non-exclusive potential sources. Do you have any references to justify these four potential sources?

L70-73 What I think is not clear here is that part of the maternal effect can be social learning and part of the maternal effect can also be (indirect) genetic inheritance. L82-83 (e.g. maternal learning) . Maternal learning might be one of the reasons of similarity between offspring and mothers.

L88-90 Multigenerational pedigree is not required if you do an experimental approach, so I would specify that this approach is required in wild populations.

L113 Why imitation and not direct learning? and how maternal learning differs from maternal effect?

L118 Maternal effects (i.e. maternal genotype or maternal environment) should come when you first mention maternal effects.

L120-121 I don't think that genetic and environmental differences are two exclusive explanations.

Results :

I agree that the maternal learning effect can fade over time, but so can the other maternal effects, such as the genetic and environmental maternal effects. Can you explain why you made the choice to account for time since separation for maternal learning but not for the other maternal effects (random effect part)? Ignoring time for the maternal effect could, in part, bias your estimate of heritability downward. In addition, given that you have motherID and pedigree, I was wondering if there was any particular reason not to partition the maternal effect variance into genetic and environmental maternal effect.

I did not understand why the time x maternal trophic position was considered as maternal learning? You show in L119-137 of the supplementary material that the distance between offspring and mother increases with time. So your interaction time

by maternal position effect might just be a spatial effect due to dispersal and not due to learning fading over time.

L162- You said 72 females in the introduction.

L226-229 The legend labels need to be more precise: environment -> environmental similarity to be consistent with the text legend; permanent individual -> permanent environment (to be more consistent with quantitative genetic literature?). Heritability is a ratio and not the variance components of interest, so something like "direct additive genetic" would be better. Please make sure this figure is color blind friendly (same for figure 3).

Discussion

The authors mainly discuss the results of maternal learning, which gives the feeling that they only studied this effect, which is not the case. There are other components that are interesting and have been poorly studied in the literature.

L257 The 8% permanent effect can also be due to a cohort effect and many other things.

L262 Sometimes you have distinguished social and maternal learning, sometimes you have used them as synonyms.

L289- But if there is social learning, do you think that removing one problematic individual from the population will be enough?

Methods

I think this section also needs more guidance for the reader. First, it is not clear on what number of individuals you did the quantitative genetic analysis. Then I think it would be nice to have the model equation to better understand all the effects that were tested. You have a lot of different variance components and it is not clear what all your components together represent. For example: permanent environment is between individual differences not explained by additive genetic, maternal effect, habitat similarity, etc. For the maternal learning vs. maternal effect variance components, you could perhaps specify that maternal effect it is the rest of the maternal effect that is not learning. Finally, it might be good to specify what percentage of females disperse and at what distance, so that we can better understand the biological conditions under which we can disentangle environmental similarity from other components.

L399- Please provide quick pedigree summary statistics (such as the number of generations).

L433 It would be good to know the average home range size of the bears so that we can find out that 25 m spatial resolution is good.

L458-459-472 You should add that you calculated the relative importance of each variance component and the associated credible interval, since these are the values you provided in the text. It might also be nice to provide the estimated variance of the full model in the supplementary materials.

Supplementary

L78-79 : Why is it written "among individual variation explains all the variance in trophic position" when the R^2 is 45% ?

L148-147 : I don't understand why the spatial distance approach is better for this particular point (since it gives no information about ant or moose density). Individuals close in space may have similar phenotypes, but you will not know if this is due to similar moose/ant densities.

L156-157 Similar trophic positions due to social learning or shared genes.

L170-171 It will be worthwhile to provide the autocorrelation coefficients by distance class so that we can understand at what spatial scale your habitat composition is autocorrelated and whether it's less than the dispersal distance.

L184 Why 98% and not 89% as in the main text?

Reviewer #3

(Remarks to the Author)

I appreciate the attempts of the authors to pull together this information across such a long-term dataset. In revisions, it would be helpful to focus on several specific points to assist the reader in determining the validity of your work:

1. Lines 123-159 in the Introduction reads more like Methods than Introduction material. The authors even include references to Fig. 1A and Supplemental material before any results are included. I would suggest the authors include readily defined hypothesis, objectives, or predictions then use those for organizing the methods and results.

2. The amount of details on stable nitrogen isotopes timing in hair (Lines 126-131) appears to be Methods that are placed into the objectives or it could be introductory material that might be better in a separate paragraph before your objectives. It appears that the majority of your analysis is based on accumulation of diet in hair so perhaps including these details in a separate paragraph, and adding more detail on how it has been done in bear or other species, would set up your final paragraph better than how it is currently constructed using Fig. 1?

3. Based on 1 above, it is not clear in the Methods, which of these methods were needed for each objective because each objective is loosely presented.

4. Related to 3 above, if hair is from "known" mother-daughter pairs (Line 132), and the methods in the Pedigree section were from Frank et al. , then there is no need to include this in your methods. I would suggest you state that mother-daughter pairs were determined in Frank et al. and not include the paragraph Genetic Pedigree and parentage assignment in the Methods because that component was already completed prior to this effort.

Fig. 1A: It is not clear to me the need or intent for this figure? It seems citations confirming the relationship of diet to hair and hair growth across bear and different species as suggested above would be more valuable than a figure of this nature. In

addition, the text cites Jimbo et al. and Cattet et al. for Fig. 1A so was this created and presented in those publications? If so, it should not be included here. If it was created in this effort then it should be in Methods and a Result of this work and not in the Introduction.

Some specific comments by line below with the above points in mind:

Methods

Lines 367-370: The paragraph states that natural foods, plural, were collected but only moose hair was collected. Why was only moose hair and not muscle collected considering bear likely are not eating a great deal of hair? Would the authors expect moose hair and muscle to be similar in nitrogen stable isotopes or were both measured to confirm? It is not clear to me that hair is a good measure or comparable measure of muscle or reflective of stable nitrogen isotopes that would be incorporated into bear hair.

It appears that other food items were collected (Supplemental Figure S1) but only moose were referenced here in the methods?

Line 406: Citation 61 is not complete so please include additional details.

Line 411: Is Supplementary analysis 3 different than Supplemental Figure S3? If not, terms should be consistent.

Figure S2 and S4 are not referenced in the manuscript.

Environmental Similarity: There are several clarifications needed with this data that I believe the authors should consider:

1. What is the sample size for these GPS and VHF collars that had >1000 GPS or VHF locations? It is difficult to assess if every bear that hair collected during capture was analyzed (so sample size reflected in first line of the Results) or if the Environmental similarity was based on a subsample of this assuming GPS/VHF collars failed, bears died, etc., thus a smaller sample size than what was originally collared and had hair collected. Line 431-432 is not adequate to inform the reader of the sample size that this entire section and analysis was based.

2. 95% kernel density estimator – This estimator is lacking considerable information considering it was done with both autocorrelated GPS data and likely not autocorrelated data with VHF telemetry. The program used for the kernel density estimator should be included as well as the smoothing option would be a minimum amount of information needed here. It would also be helpful to understand the reason for this choice if it was to “extract habitat composition in each bear’s lifetime home range.” Do the authors believe this is the best choice of an estimator considering it has been documented to be one of these least representative estimators of home range across many species?

3. Extracting home range composition for the “median” year if multiple years were monitored needs additional details. This might be appropriate for bears monitored 1-3 years but if monitored 5-10 years I question the reliability of the Corine Landcover Maps specifically if they are static and collected/released annually or in 5 year intervals? The lack of details on this data source makes this difficult to evaluate.

Reviewer #4

(Remarks to the Author)

The authors used a very impressive dataset with repeated individual annual estimates of bear trophic position (derived from stable isotope analysis of hairs) to study individual dietary specialization. They partitioned among-individual variance in trophic position to discern the relative contributions of social learning, genetic predisposition, environmental forcings, and maternal effects. They conclude that social learning and maternal effects are as important for individual dietary specialization as environment. This result is very interesting because individual dietary specialization is key for species’ resilience against environmental change. It is also very interesting given the paucity of such detailed analyses in the literature.

I thus agree with the authors that the question and the results are important, and had no issue with the analyses. I was also very impressed by the gigantic amount of work behind this study, and found enough details in the methods for the work to be reproduced (at least in principle).

I do have three main criticisms, however; 1) lack of discussion regarding the pertinence of trophic position to evaluate dietary specialization, 2) apparent mistakes in some of the numbers that are presented (sorry if I am wrong on that point), and 3) need to clarify the text. Below I detail each point or give specific examples.

TROPHIC POSITION VERSUS DIETARY SPECIALIZATION

A given trophic position could be obtained from various diets and from various levels of dietary specialization. For example, a protein-rich diet could be obtained from various proportions of predation versus scavenging, or from various proportions of moose versus ants. Similarly, a rather generalist bear (e.g., feeding on crowberry and some ants) could have a similar trophic position as a specialist bear feeding exclusively on bilberry. I don’t know enough the ecology of brown bears to evaluate whether these examples are realistic, but I am questioning the fact that trophic position is used to investigate

dietary specialization without any discussion on this methodological issue.

APPARENT MISTAKES IN SOME NUMBERS

72 bears (line 91) versus 71 bears (line 162)? There is perhaps a good reason for this difference, but I did not find it.

Environmental similarity accounted for 9% of phenotypic variation in trophic position (line 174) versus 12% (line 45)?

Line 174 – “9% [0.1 - 5%]”. I don’t see how the 89% credible interval can be [0.1 - 5%] given a median of 9%.

I tried with no success to match the percentages given in Results with the width of the color bands in Fig. 2.

NEED TO CLARIFY THE TEXT

Please always use the same term to name a given source of variation, except when defining that source of variation (in which case synonyms can be useful). For example, genetic predisposition (line 41), genetic heritability (line 46), genetic inheritance (line 71), and genetic relatedness (line 134) seem to be used interchangeably, which generates lots of confusion.

Fig. 1B and Fig. S3C. Can’t you give a name and some units to the Y axis?

The legend of Fig. 2 would be easier to read if variables were given in the same order as they appear on the figure, which ideally should be in the same order as they appear in Results. Please also better explain the two rows of the figure (Basic and Maternal learning) in the legend of the figure. In short, whereas the first paragraph of Results, and Fig. 2, give most of the Take Home message of the paper, they lack the clarity that they deserve.

Fig. 2 – can you indicate the percentages of variance directly on the figure, as this will help relate the figure to the text (for example, indicate 11% inside the blue box).

ADDITIONAL COMMENT

Line 257. You indicate that “variation solely linked to individual variation (in our study 8 %) demonstrates potential for behavioral innovation and the potential to adapt to changing conditions”. Can you better explain this proposition?

Version 1:

Reviewer comments:

Reviewer #1

(Remarks to the Author)

The revised manuscript flows well and reads well. Points were clarified well. I have a few extremely minor comments/suggestions that I hope will further improve the manuscript.

L209-L211: I found this statement a bit vague. "siblings occupying consistently lower or higher" - does this mean that sibling A was always higher than the daughter and sibling B always lower (for example)? How did this translate to different individuals?

L275: I suggest a slight rephrase: "were not significant," rather than "were insignificant".

L288: I suggest "associated with" rather than "associated to".

L403: Can you please clarify that hair samples includes bear samples and moose samples?

L472 and 475: These should be plural: generations and reconstructions.

(Remarks on code availability)

I accessed the data, but did not run the code. The provided PDF under "learning of carnivory" demonstrated how the code was used. I did not feel that I could review the statistics in detail as I am not that familiar with some of the analyses. However, I think that the methods are reproducible and could be run using the information provided.

Reviewer #2

(Remarks to the Author)

I was also a reviewer of a previous version and I appreciate the effort made by the authors to respond to the four reviewers. The authors are running complex quantitative genetic models, attempting to take into account various effects, which is commendable. However, I still have concerns about the 'social learning' effect in this complex model.

I am concerned about the maternal trophic position fixed effect (also known as the social learning effect) and its interpretation. I agree that the hybrid approach you are using proposes to fit the maternal traits 'x' as a fixed effect, but most of the time, this is not done with the exact same trait between offspring and mother. Correct me if I'm wrong, but the model appears to be: Cub trophic position ~ maternal trophic position * time since separation + age + (1|ID) + (1|maternal ID) + (1|environmental similarity). Here, because the maternal and offspring phenotypic traits are the same, your 'social learning'

effect comprises both additive genetic and maternal effects. In other words, you are investigating the genetic basis of trophic position while using maternal trophic position as a fixed effect, which itself has a genetic basis and is highly genetically correlated with offspring trophic position because it is the same trait.

The consequence of this is that you are 1) underestimating your additive genetic variance and thus heritability, and 2) discussing how social learning is important while the estimated percentage might be highly inflated. One way to quickly assess if this is a significant issue would be to run the same model without the 'social learning effect' and observe how it affects your variance components. However, please refer to Pick et al., 2019 (DOI: 10.1002/evl3.125) where they encountered a similar issue and applied additional equations to correct the problem described above.

Additionally, considering the interaction between the 'social learning' effect and time since separation, I find the interpretation of VSL complicated. This interaction implies that the percentage of VSL and heritability can change over time, a point which you later discuss in the manuscript. However, how did you accommodate the interaction between 'social learning' and time since separation when calculating the variance attributed to social learning? Was the percentage of VSL estimated for a median time since separation?

Finally, to address your question regarding the partitioning of maternal genetic and permanent environmental effects, in the reference you cited (McAdam & colleagues), they stated: 'Given a pedigree structure with at least three generations' depth, the animal model can be further extended to partition VM into genetic (VMG) and environmental (VME) components, and to estimate the covariance between direct additive and maternal genetic effects (COVAMG).' In practical terms, this implies fitting an additional random effect with maternal identity linked to the pedigree, similar to what you do for individual identity to distinguish between additive genetic and permanent environmental effects. However, considering that your model already has to distinguish between many different components with a low sample size, I no longer believe it is advisable to add any additional effects to your model.

Point-by-point comments

L49: Please also provide the value for heritability.

L217-219: Partly confounded.

L287: Between-individual effects include both genetic and permanent environmental effects, so please specify 'permanent between-individual effects.'

L290: Are you referring to learning from sources other than the mother? I am not sure I understand what you mean by 'differential learning.'

L292: The term 'adapt' suggests an evolutionary response; perhaps 'adjust' would be more appropriate here.

L322-324: Considering that a large percentage of phenotypic variation is attributed to environmental similarity, don't you think that changing the relocating bears of their habitats (translocation) could be a solution? If bears are plastic enough, their trophic position might be influenced by changes of the environment.

L354: Does this mean you consider food resource similarity to be different from habitat similarity? How do you define habitat similarity ?

L488: How can you be certain that age is not confounded with among-individual effects?

L500: Please specify the total number of observations for the animal model. Is it 213 ? because in the Table S1 you have much more observations (n=335).

L506: If you partition V_i into its sources, V_i can't be cited as one of the sources! From what I've read, I understand $V_i = V_{age} + V_{VSL} + V_i + V_E + V_a + V_M + V_R$.

L516-517: If age and time since separation are perfectly correlated, why do you include both as fixed effects?

L522: Time since separation and age are the same effect? Why use two different names for it? I am confused.

L529-530: With a Gaussian distribution family?

L335 of the supplementary: Please provide the total number of observations in your supplementary material; repeated measurements are known to increase statistical power. Also isn't it 71 individuals?

Table S1: Why is it listed as 95% CI while you have 89% CI in your manuscript?

Supplementary L355-356 : You find a pvalue of 0,07 so if I understand well it means that 7 % of the permuted models had higher additive genetic variance than the model with observed dataset. Which means that in 93 % of case you have more genetic variance than by chance and thus genetic variance in trophic position exist, or maybe I have misunderstood ? (« if related individuals have similar dietary specialization, our observed pedigree should explain more variance than a permuted pedigree). But you conclude that you did not find any evidence of heritability because you have a CI overlapping 0,1 % while

here your permutation results seems to indicate that you do have an genetic variance (and heritability) but that you are not able to quantify it accurately. Also what does a p-value of 0,07 mean in term of power ? In Pick et al 2023 they defined power as « the proportion of 'observed' datasets in which the p-value was below a nominal threshold of 0.05. » So I don't see how it reflect power in your specific case.

Another way to deal with « statistical power » is to use the head size for which you previously found moderate heritability (10.1111/eva.12786, $h^2=0,24$ [0,06-0,38]) and subsample your dataset to calculate new heritability with 71 individuals and XXX observations. So if you don't find any heritability or really low heritability you can be sure that your model struggle to estimate genetic variance and that it doesn't mean there is no heritability (but also see my comment about your maternal fixed effect that might under estimates your V_a).

(Remarks on code availability)

Reviewer #3

(Remarks to the Author)

I agree with the other reviewers that this was an impressive dataset, analysis, and compilation of data, however, that is also where it is deficient in so many ways. The authors argue for individual diets of bears such that Lines 114-117: "Populations range from tracking food resource pulses, such as spawning fish, scavenging on ungulate carcasses or preying on ungulate calves, or feeding extensively on invertebrates, to populations using primarily fruiting plant-based diets." With this breadth of diet, we would expect a simple analysis to document this which would include a figure like Figure S1 of all bears in their dataset. Presumably, bear foraging on carcasses would be different from those foraging on fruiting plants are at least separation of bear pairs that exhibit different overall diet selection.

The authors dismissed my comments on moose hair and basically stated that moose hair is equal in isotopes to moose meat or tissue consumed by bear. I realize that bear consume hair with the carcass but bear are not digesting and assimilating hair as they are moose tissue. As stated by the authors, bear and moose hair reflects a diet over time whereas tissue is within weeks of consumption and assimilation in the bear. It is not clear how moose hair can be a proxy for diet of moose?

Also, using KDE home ranges with reference bandwidth smoothing likely severely over-estimates the size of the area used by each bear. Even if we overlook the issue of GPS versus VHF telemetry in the sample size (Bears were monitored for a minimum of 1000 GPS locations ($n = 47$) or were located via VHF on at least 25 days ($n = 24$) so comparing home ranges of 1,000 to 25 locations as being equal, but centroids to core home ranges for home ranges that are considerably larger than what the bear uses makes this analyses difficult to interpret and justify. The authors do not provide this raw data, shapes of the home ranges used to determine overlap/distance between core areas, or how using 1000 versus 25 locations depending on bear would influence their results. Even in their rebuttal to my comments on this, a sentence is cut off mid way.

The authors also dismissed my comment that figure 1A is not needed because it just repeats text and provides no further understanding of bear molting timeframe. I also have no idea what value there is in Figure 1B but if it is from a separate dataset, I don't see why it is included.

How does the annual changes in home range habitat composition being negligible (Lines 456-458) show up in Figure S5 that focuses on distances between natal range centroids?

(Remarks on code availability)

The code is mostly output only code. Nobody with a comparable dataset could replicated their research and we are not even able to replicated what they assessed with so much data missing. There is no raw data to actually perform any analyses in a reproducible way. The authors saved .rds files after running analyses on raw data thus we can only evaluate what the authors chose to provide. They included code for reproducing figures that are already included in the manuscript which seems to have limited value. Specifically, there are no bear telemetry locations to reproduce the home range estimates. They included bear C13 and N15 for 71 bears but some females had 1-7 daughters and appear to be considered independent or separate samples.

I understand if the other reviewers provide a more useful review for the journal but figured I would provide what I could.

Reviewer #4

(Remarks to the Author)

The questions I had raised during my initial review have been answered with satisfaction, the needs for clarification have been taken into account, and the mistakes I had pointed out have been corrected.

I am still impressed by the gigantic amount of work behind this study and have no doubt it represents a significant advance in the field. I congratulate the team for their research.

(Remarks on code availability)

Version 2:

Reviewer comments:

Reviewer #3

(Remarks to the Author)

1. There is no point to Figure 1A considering it is a diagram of the exact sentence (Lines 150-151) stating: Bear hair is annually renewed through molting in June, regrows over the summer and fall and stops growing during winter hibernation.

2. Stable isotopes of moose hair being used when the authors acknowledge that a majority of diet is moose meat but moose meat was not available for stable isotope assessment. The authors then cite Stephens et al. (2023) that "arithmetic correction to convert hair to meat" was 4.0 per mil for hair and 3.9 per mil for meat so there is no difference between the two. My concern is that correction factors aside, we simply don't know. What we do know, and what was confirmed by the authors, was that bear don't consume moose hair but they do consume moose meat.

3. Size of home range - The authors claim that: "In fact, KDE is known to underestimate home range sizes, not overestimate." The authors should be required to show supporting citations to this statement but they are completely wrong. The authors did investigate another KDE estimator and found the centroid distances shifted or distances increased for some bear, again supporting that KDEs to overestimate sizes of home range.. Trading out one KDE estimator (reference bandwidth smoothing) for another (AKDE) does not really address my comments.

4. Code and data availability - the authors state in their rebuttal: "Providing raw data as rds files is common standard. This file format is advantageous over other data formats as it is more stable (files cannot be opened and cell contents changed, no problems with decimal delimiter). Our final dataset indeed includes a column of model derived data, i.e., maternal trophic position. Code to reproduce this column and to derive this final dataset is provided in the Supplementary Code S1-S3. "

This code and most data was not previously provided, I have it downloaded because I looked into this in the last revision. Some data is now available as .rds only because we instructed that it be made available.

(Remarks on code availability)

This code and most data was not previously provided, I have it downloaded because I looked into this in the last revision. Again, the authors dismissed my comments that it was provided and I simply missed it. Some data is now available as .rds only because we instructed that it be made available. HOWEVER, this is not data. This is only output of analysis with some raw data. We can't generate home range isopleths from their code (the authors acknowledge they can't share GPS/VHF locations of bear) but they also don't give us shapes or sizes of resulting polygons from their home range analysis. The data they provide is equivalent to providing results in Tables in the text or Supplements so provides no value towards reproducible science in any way.

In conclusion, the authors provide considerable data of results and to make figures which is not the purpose for open science. I understand that the authors may not be able to share actual bear locations, but I am not able to determine shapes or sizes of home range and not able to assess their habitat composition without raw polygons of home ranges either. The data and code to run models is provided as confirmed by the authors so the analysis is reproducible fo that portion of the manuscript.

Version 3:

Reviewer comments:

Reviewer #4

(Remarks to the Author)

(Remarks on code availability)

REVIEWER COMMENTS

Reviewer #1 (Remarks to the Author):

The paper was well written, and the results were interesting. The value of having this larger dataset was quite clear. I did have some queries that are mostly seeking clarity but suggest that these will improve the manuscript further.

Thank you for your overall positive evaluation of our work. We appreciate your time and input and hope to have addressed your comments to your satisfaction. We also appreciate your thorough language editing.

Abstract:

L37: The apostrophe after “species” is not really necessary.

Changed

L39: Add a comma after “drivers”. Use the plural of remain (remains).

Added. As “remain” relates to “associated underlying social learning, genetic, and environmental drivers”, i.e. plural, kept as is.

L42: After reading the manuscript, I’m not entirely clear on what you mean by “environmental forcings”. As you only use this term in the abstract (here and L50), I suggest using more relevant terminology that matches the main text.

Changed to “composition”

L46-47: Rephrase: “the trophic position of offspring closely”.

Changed

L49: Replace “reveals” with “shows”.

Changed

Introduction:

My primary concern with the introduction is how maternal learning is differentiated from maternal effects, offspring learning, environmental effects and genetic effects. It seems to be blended together, and the definitions do not provide much clarity. I suggest using “offspring learning” rather than “maternal learning” if this is what you really mean (as suggested on L114). But this needs to be clearly differentiated from maternal dietary effects (e.g. offspring gaining information about food resources during suckling).

We hope our introduction now more clearly differentiates the different sources we are evaluating.

What we mean is that offspring forage together with the mother during their first year of life (they exclusively suckle while in the den but start eating solid food alongside their mother upon den emergence) and thereby learn dietary preferences from the mother by foraging with her. This should lead to a similar trophic position between mother and offspring even after separation. We originally used the term “maternal learning” to refer to social learning from the mother during the period of mother–offspring association.

Regarding the terms “offspring learning” versus “maternal learning” – we think that “offspring learning” could be misunderstood as an umbrella term for both social and individual learning, while maternal learning clearly differentiates the source of the information. The mechanism that we here propose is that “social learning” (by the offspring from the mother) leads to individual differences in dietary niche. We therefore decided to change our terminology to “social learning” throughout.

Ultimately, we agree that spatial, genetic, maternal and social learning effects can be confounded – to test which one drives the observed individual variation, one can use leave one out methods, as we have done in the supplementary material S6 and S7, and as has been shown for example in “Gervais L, et al. Quantifying heritability and estimating evolutionary potential in the wild when individuals that share genes also share environments. *Journal of Animal Ecology* 91, 1239-1250 (2022).”

We have rewritten the introduction paragraph in the following way, line 74:

“In the fields of behavioral and evolutionary biology, individual variation is measured as the variance attributed to permanent between-individual differences, while the sources of variation can be quantified using complex hierarchical models (e.g., “animal model”)⁵. In principle, three sources of variation are commonly considered: 5: variation in the environment 6, additive genetic effects from which trait heritability can be estimated 7, 8, and parental (especially maternal) effects 9, 10. In addition, individual variation can be maintained through social learning during ontogeny¹¹, an aspect that, to our knowledge, has rarely been

integrated into animal models (but see 12, 13). Here, we provide the first study to attribute individual variation in dietary niche to its sources.

Differences in the environment, in terms of habitat composition and associated availability of particular food resources, are generally considered the main cause of individual variation in dietary specialization 14. This is particularly true in range-resident species, where individuals occupy a subset of the population's range and individual home ranges vary in resource availability 15. However, beyond the environment, resource preferences have been suggested to be genetically heritable and determined through genes inherited from both mother and father, where more closely related individuals share more similar diets than distantly related individuals 14, 16. Additionally, parental phenotypes may also affect offspring phenotypes in ways other than genetic heritability 9, 10. Maternal effects are more commonly studied because mothers often have unilateral control over offspring development 9, especially in mammals, however, paternal effects are plausible in species with paternal care. Maternal effects on offspring behavior have been suggested to be lifelong, they have either a genetic or environmental basis and summarize the cumulative influence of many different proximate maternal effects, include pathways such as provisioning rates, milk production, in-utero hormone transfer, and epigenetics 9. Statistically, maternal effects account for similarities in dietary niche among offspring of the same mother (fitted as a random intercept for mother identity) 12, 13, but not for the similarity of dietary niche between mother and offspring. The latter would be an example where the maternal trait affects the offspring's trait, which statistically can be clearly differentiated from other maternal effects 12, 13. Similarities between the dietary phenotypes of mothers and their offspring indicate social learning of resource preference or competence to secure a resource by the offspring from the mother during early ontogeny 17, 18, 19, 20, 21. Social learning is therefore an additional pathway by which individual variation can be maintained. It is reasonable to assume that the effects of social learning during rearing will weaken later in life through individual experiential learning 22."

L61: Replace "hence" with "likely".

Changed

L62: Add a comma after "composition".

Changed

L63: Rephrase: "key for driving species resilience in response to"

Changed

L64: A comma may be useful after "world".

Changed

L66 onwards: This paragraph was very long and incorporated many different ideas. The concept of environmental effects wasn't well developed in contrast to the other ideas. The issue with separating out maternal learning and maternal effects is that mothers can still affect offspring postnatally because of the environment they might be experiencing independently of their offspring. I suggest breaking this paragraph up and systematically discussing each topic clearly and unambiguously.

Please see reply to main comments above, we have rewritten this paragraph.

L66: The comma after "opportunity" is not necessary.

Changed

L68: Replace "yet" with "however".

Changed

L69: Bring "been" after "rarely".

Changed

L71: This should simply be parental effects. Fathers can also affect offspring development and behaviour even in species without paternal care. I suggest introducing the concept broadly, and then discussing maternal effects specifically within that context.

We have included paternal care (lines 90), but also would argue that in species without paternal care (which is the case for brown bears), the paternal effect is limited to genetic effects. We do not see a biological plausible pathway for species with strict maternal care how dietary specialization of offspring could be determined through, e.g., paternal environment. This would be different in species with shared or strictly paternal care, where paternal effects could manifest, e.g., via paternal differences in provisioning rates, territory quality or else. Especially in mammals, there are more pathways for maternal effects to manifest: hormones during pregnancy, milk quantity and quality. For these reasons, testing for maternal effects in quantitative genetics studies is common (The adaptive significance of maternal effects: [https://doi.org/10.1016/S0169-5347\(98\)01472-4](https://doi.org/10.1016/S0169-5347(98)01472-4); Maternal effects and evolution at ecological time-scales: <https://doi.org/10.1111/j.1365-2435.2007.01246.x>; A mother's legacy: the strength of maternal effects in animal populations: <https://doi.org/10.1111/ele.13351>) while paternal effects are rarely mentioned or studied.

In summary, we briefly mention parental and paternal effects as suggested, line 90:

"Additionally, parental phenotypes may also affect offspring phenotypes in ways other than genetic heritability^{1,2}. Maternal effects are more commonly studied because mothers often have unilateral control over offspring development², especially in mammals, however, paternal effects are plausible in species with paternal care."

L74: Rephrase: "between the dietary phenotypes of mothers and their offspring".

Changed

L75: Replace "are" with "may be".

Changed

L78-79: Again, paternal effects may also be important. As suggested, restructuring this paragraph and then focusing on parental effects specifically will allow you to introduce the concept broadly and then refine it to explain why maternal effects are a potentially stronger factor for solitary species.

As described above, we do not see a direct pathway how paternal effects (other than genetic effects) could manifest in species with strictly maternal care.

L81: Please provide a reference for this statement. It is not clear why maternal effects would be evident from the similarity of repeated samples of siblings. Couldn't a genetic effect equally result in a similar response between siblings?

Statistically, maternal effects are tested by fitting a random intercept for mother identity. This introduces a "grouping factor" for all samples of offspring with the same motherID and tests whether maternal identity explains variance.

We have revised the sentence in the following way and added references, line 97:

"Statistically, maternal effects account for similarities in dietary niche among offspring of the same mother (fitted as a random intercept for mother identity) 12, 13, but not for the similarity of dietary niche between mother and offspring. The latter would be an example where the maternal trait affects the offspring's trait, which statistically can be clearly differentiated from other maternal effects 12, 13. Similarities between the dietary phenotypes of mothers and their offspring indicate social learning of resource preference or competence to secure a resource by the offspring from the mother during early ontogeny 17, 18, 19, 20, 21."

The genetic pedigree in contrast creates a similarity matrix among all individuals in the population, where the genetic difference of each individual to all other individuals in the population is calculated and tests whether more closely related individuals are also more similar in their dietary specialization.

Although indeed, offspring of the same mother and father are closest related "genetic heritability" also includes other links such as half siblings, cousins, and aunts or uncles, line 87:

"However, beyond the environment, resource preferences have been suggested to be genetically heritable and determined through genes inherited from both mother and father, where more closely related individuals share more similar diets than distantly related individuals^{3,4}."

L82-83: Likewise, please provide a reference for this statement. It is not clear why similarity between mothers and offspring reflects maternal learning (which should be defined here because it's not clear) when similarity could be a consequence of genetic effects, maternal effects or offspring learning from their mothers.

We hope that it is now clearer that "social learning" signifies offspring learning dietary preferences from their mother during rearing, see response above.

L94: Add a comma before "we then attributed".

Changed

L99: I suggest "newborn ungulates" rather than "ungulates neonates".

Changed to "ungulate calves"

L100: Hyphenate "plant-based".

Changed

L101: Delete the comma before "however" and replace with a semi-colon.

Changed

L106: Use plural "sources".

Changed to: "The most parsimonious source of variation in diet is, therefore, heterogeneity in the environment."

L107: I suggest starting a new paragraph before "Brown bears".

Changed

L110: Add a comma after "mother". Replace "assume" with "predict".

Changed

L112: Add "may" after "hence". Use plural "behaviors".

Changed

L113: Delete "hence". Replace "of" with "from".

Changed but retained "of"

L114: Delete the comma before "however" and replace with a semi-colon.

Changed

L116: But body size is also strongly influenced by environmental effects, such as resource availability, so could equally affect maternal effects through effects on maternal environment.

Agreed – body size is genetically heritable (24%) but is also driven largely by environmental effects (50%, Rivrud et al 2019). We here wanted to give an example that heritability has been shown for this population. We have rewritten the sentence in the following way, line 126:

"For example, while body size is determined largely by resource availability in the environment, it has also been shown to be genetically heritable in our study population 30, suggesting greater similarity among closely related individuals also in other linked traits, such as trophic position."

That is true,

L117: Add a comma after "population".

Changed

L121: With reference to “leading to greater similarity” - This assumes no competition between siblings (either in utero or ex utero). Runts could be argued to gain fewer maternal resources, so why would you expect similar trophic positions if there are differences?

Response: Indeed, sibling competition is possible and could affect the suggested pathway by which maternal effects could manifest (differences in maternal milk quality → differences in offspring body size among mothers → differences in offspring trophic position among mothers). The demonstrated maternal effect would then be caused by other pathways. See revised line 94:

“Maternal effects on offspring behavior have been suggested to be lifelong, they have either a genetic or environmental basis and summarize the cumulative influence of many different proximate maternal effects, include pathways such as provisioning rates, milk production, in-utero hormone transfer, and epigenetics 9.”

L123: I suggest starting a new paragraph before “To assess”.

Changed

L128: Delete the comma after “may”.

Changed

L132: Past tense: “fitted”.

Changed

L133: Consider “untangle” rather than “disentangle”.

Changed to “separate”

L134: Delete “as determinants of” and replace with “to”.

Not changed as we considered the sentence clearer in its original form.

L136: Add a comma after “ranges”.

Changed

L137 and elsewhere: I suggest putting all the “such as” parts in brackets as examples. “Mature habitat (e.g. old and mid-successional forests)” – and so on.

Changed here

L142: Should this read: “a mother’s trophic position on her daughters’ trophic positions”?

Indeed, changed

L144: Why does 3 come first? Consider reordering to have the material presented in numerical order.

We agree that this is confusing, the logical order of the methods presented in the supplements is however that steps S1-S2 are prerequisites before S3. These are referenced in the Methods, however, given that the Methods are presented at the end of the manuscript this creates a discontinuous referencing to the supplements.

L146: Replace “last” with “Finally”.

Changed

L150: Why is this in supplementary material only and not in the main text? Perhaps indicate here why you provide this information as supplementary, considering you make the point that you are focusing on daughters. **We think that every additional analysis provides more information and support for our conclusions. In reality, the most solid part of the paper - a model that accounts for confounding sources – could only be fit on female offspring because we have only limited data on male offspring after independence and we would not be able to trace the effect of maternal learning over time. While it is sure an interesting observation that males are equally similar to their mothers in their first years of life this analysis neither accounted for relatedness, the environment or maternal effects, it was based on a simple correlation. Given the journal space constraints, we decided to present this information in the supplementary material and to focus on the most sound and complete part of our study in the main texts. We have changed the sentence here to (line 169):**

“As we were interested in lifelong variation of dietary niche, and male offspring were only monitored for a short period after family breakup, we focused on individual specialization of female offspring.”

L156: Delete “last”.

Changed

Results:

These were mostly easy to follow.

L162 Replace “in” with “from”.

Changed

L164: Replace “revealed” with “showed”.

Changed

L177: Delete “as”.

Changed

L179: But in L176 you say that siblings were more similar in trophic position. So how do you know this is a maternal effect and not a genetic effect? Please elaborate.

The genetic relatedness matrix is distinct from the maternal effect in two aspects: a) it takes into account maternal and paternal relatedness pathways (e.g. paternal half siblings are as closely related as maternal half siblings) while the maternal effect is limited to maternal pathway, b) genetic relatedness is a

spectrum including maternal, paternal, full sibling, half-sibling, aunt or cousin relatedness while the maternal effect is limited exclusive vertical transmission from the mother to her offspring. Elsewhere, not accounting for maternal effects has been shown to upward bias estimates of heritability (White 2019) and the authors advised to control for both if possible. Statistically we can test this by leaving one term out and monitoring whether the variance explained by the remaining term increases. We did this in the supplementary analysis and did not find that genetic heritability was biased by maternal effects. Given that the maternal effect does come out as important, other maternal attributes than genes, such as her home range, size, or milk production, seem to play a role in explaining similarities among maternal full- and half siblings.

We have added the above argument in brief to the sentence (line 200):

“Genetically more closely related individuals (including paternal half-siblings, aunts, uncles or cousins) did not share a more similar trophic position (2% [5%, <0.1% – 17%] of variance explained), providing no evidence that dietary specialization could be heritable in this population (Fig 2, see S11 for the full summary table).”

L183: Replace “as” with “to”.

Changed

L193: Start the point after the colon with a small letter “a”.

Changed

L194: Commas before and after “but not heritability” could increase the clarity and flow of this sentence.

Changed

L195-196: Perhaps I missed something here, but I don’t understand how controlling for spatial proximity showed that it was a confounding effect. Perhaps consider additional elaboration for clarity here.

We have rephrased this sentence as follows (line 217):

“However, spatial distance and maternal effects seemed to be confounded in this female philopatric species (S7): after excluding spatial distance, maternal learning and maternal effects, but not heritability, explained more variance in trophic position, indicating that spatial proximity and maternal effects are confounded because settlement home ranges of philopatric daughters are often close in space to their mothers forming so called matrilineal assemblages.”

L201: Rephrase: “offspring trophic position”.

Changed

Discussion:

L246: Rephrase: “between the trophic position of mothers and daughters slowly”.

Changed

L251: Delete the comma before “however” and replace with a semi-colon, adding a comma after “however”.

Changed

L253: Using the term “early-life imitation” here is confusing. As suggested earlier, I suggest using a single term, offspring learning, in place of others (including maternal learning) so that the reader isn’t left confused in the discussion.

Changed to “early-life social learning of maternal dietary preferences”

L258: Replace “adapt” with “respond flexibly” as this is likely a better reflection of what is occurring, rather than adapting (in the classic evolutionary sense).

Changed, we agree that adapt sounds to “evolutionarily

L275: Delete the comma after “parental care”.

Changed

L278: Singular “remain”.

Changed “but the link to parental learning of foraging preferences remains unclear”

L282: Delete “last” and replace with “finally”.

Changed

L293: Rephrase: “over their lifetimes, further indicating the crucial”.

Changed to “over their lifetimes, remarking indicating the crucial”

L295: Add a comma after “resources”.

Changed

L297: Delete the comma after “effects”.

Changed

L300: Delete the comma after “specialization”.

Changed

L306: Consider “untangle” rather than “disentangle”.

Changed

Methods:

L369: Rephrase: “diet of brown bears in”.

Changed

L379: Please indicate what times seed preference tests were conducted.

We are unclear what the comment is referring to; seed preference tests were not part of our study.

L391: Delete "that".

Changed

L423: Consider "Furthermore" rather than "further".

Changed

L461: Delete the comma after "deviation" and replace with a semi-colon.

Changed

L473: Past tense "fitted".

Changed

Supplementary material:

It was disappointing that most of the supplementary material was not referred to in the main text to provide context. I suggest adding the citations to the supplementary material at relevant and appropriate points in the text so that the reader has context and can expand their understanding as necessary.

We were surprised by that comment as we referred to all but one supplementary content in the original version. We have now referred more often to our extensive supplementary material in our updated version.

Supplementary material S1 – line 393, 410

Supplementary material S2 – line 199, 481

Supplementary material S3 – line 164; 181, 223, 382, 477

Supplementary material S4 – line 166

Supplementary material S5 – line 450

Supplementary material S6 – line 171; 209; 459

Supplementary material S7 – line 173; 211

Supplementary material S8 – line 168; 214

Supplementary material S9 – line 169; 221

Supplementary material S10 – line 175; 224

Supplementary material S11 – line 204

Supplementary material S12 – line

Reviewer #2 (Remarks to the Author):

The authors investigated what mechanisms drive among-individual variation in diet composition (also known as individual specialization) in the European brown bear. To address this question, the authors used a complete dataset in combination with an appropriate and complex quantitative genetic approach. They showed that genetic heritability is weak and that most variations are explained by maternal effects (including maternal learning). To the best of my knowledge, this is the first study to estimate the heritability of dietary composition and to consider the many different confounding mechanisms (habitat similarity, maternal effect, permanent environment) at the same time. These results are very promising because they involve a species that can come into conflict with human populations due to the type of dietary specialization of certain individuals.

While I am convinced that this paper will move the field forward, I think it should benefit from some clarifications in the introduction and methods to help the reader through the paper. In addition, the authors have made an effort to account for a large number of confounding effects, but I think a first step would be to include a statistical power analysis (in the Supplement) to understand what the chances are of detecting a true heritability with a basic animal model.

Thank you for your overall positive evaluation of our work. We have tried to address your comment regarding a power analysis in the supplementary material.

In essence, a power analysis reveals the probability to reject the null hypothesis (i.e., to detect that an effect is statistically significantly different from 0 at a prescribed alpha level, usually 0.05) given the sample size.

We see two problems with implementing a power analysis on variance components in a Bayesian approach: a) in a Bayesian approach inference is already made on the “posterior distribution” and not based on null hypothesis testing with an alpha level, and b) because variance components never overlap 0 (they can only become very small) it is not straight forward to assess significance. The power analysis therefore cannot per se answer the question whether we can detect “true heritability” because heritability is estimated on a spectrum from 0 – 1, how much variation (as a proportion of the total variance) can be attributed to the genetic effect. In our models this proportion was always very low (median between 3% – 5%, Supplementary material S7). Now there has been a recent paper on this matter “*Pick et al (2023) Describing posterior distributions of variance components: Problems and the use of null distributions to aid interpretation, Methods in Ecology and Evolution, DOI: 10.1111/2041-210X.14200*” and we followed their advice to try to address your comment.

First of all, the variance explained by heritability in our model was small, and our sample size was small which led to an asymmetrical posterior distribution of explained variance (see Figure S7). According to Pick et al. (2023), asymmetrical posteriors are generally an indication of low power. We report the median of the posterior distribution as measure of central tendency (Kruschke 2014, McElreath 2020), which also seems most robust for variance components (Pick et al. 2023).

Pick et al (2023) suggest to generate a null distribution, i.e., where no variance is explained by a given variance component, and to compare the observed and simulated model to gain inference on the significance of the effect and the power of the analysis. We have implemented their idea with a permutation. We kept the architecture of our pedigree (Dam/Sire pairs) but randomly assigned parents to offspring. If related individuals have similar dietary specialization (i.e., heritability of dietary specialization), our observed pedigree should explain more variance than a permuted pedigree. A p-value can be generated by calculating the proportion of simulated/permuted models where the explained variance is larger than the explained variance in the observed dataset.

We fitted a reduced “basic animal model”, controlling only for fixed effects, bear ID and genetic structure. We permuted our dataset 1000 times, fitting 1000 animal models as null distribution. Our observed genetic pedigree did not explain more variance than the permuted null distribution ($p = 0.077$). We have added a description to assess power to the supplementary material S12.

I have made other major and minor comments which I have grouped by sections.
Introduction:

I feel that the second paragraph needs to be rearranged to be clearer and more effective. It might not be easy to understand how the different mechanisms differ from each other. For example, it says maternal learning and maternal effect, but in my opinion maternal learning is part of maternal effect (and it is more or less what you say in the supplementary “maternal effects (both maternal learning and maternal effects)”). Regarding maternal learning mechanisms, I am confused by the use of interchangeable words between maternal imitation and

learning. Imitation is a special case of learning, but it would be better to stick to one term. Regarding genetic inheritance mechanisms, I think that maternal genetic effect is a kind of (indirect) genetic inheritance.

Thank you for your comment, we have fully revised the second paragraph of the introduction. We have tied the different sources of dietary specialization more closely to the modelling framework we use. Specifically, we are using the hybrid method suggested by “McAdam AG, Garant D, Wilson AJ. The effects of others’ genes: maternal and other indirect genetic effects. In: Quantitative Genetics in the Wild (eds Charmantier A, Garant D, Kruuk LEB). Oxford University Press (2014).” to disentangle the effect of a maternal trait on her offspring’s trait (here trophic positions which we interpret as social learning by the offspring from the mother during rearing) and other maternal effects. We have made sure to not use interchangeable words and are now strictly referring to “social learning”. Regarding maternal and genetic inheritance mechanisms, indeed the maternal effect is considered additional to additive genetic effects which are quantified using full pedigrees including paternal links, aunts, uncles, etc. Our revised introduction paragraph reads as follows, line 74:

“In the fields of behavioral and evolutionary biology, individual variation is measured as the variance attributed to permanent between-individual differences, while the sources of variation can be quantified using complex hierarchical models (e.g., “animal model”)⁵. In principle, three sources of variation are commonly considered ⁵: variation in the environment ⁶, additive genetic effects from which trait heritability can be estimated ⁷, ⁸, and parental (especially maternal) effects ⁹, ¹⁰. In addition, individual variation can be maintained through social learning during ontogeny¹¹, an aspect that, to our knowledge, has rarely been integrated into animal models (but see ¹², ¹³). Here, we provide the first study to attribute individual variation in dietary niche to its sources.

Differences in the environment, in terms of habitat composition and associated availability of particular food resources, are generally considered the main cause of individual variation in dietary specialization ¹⁴. This is particularly true in range-resident species, where individuals occupy a subset of the population’s range and individual home ranges vary in resource availability ¹⁵. However, beyond the environment, resource preferences have been suggested to be genetically heritable and determined through genes inherited from both mother and father, where more closely related individuals share more similar diets than distantly related individuals ¹⁴, ¹⁶. Additionally, parental phenotypes may also affect offspring phenotypes in ways other than genetic heritability ⁹, ¹⁰. Maternal effects are more commonly studied because mothers often have unilateral control over offspring development ⁹, especially in mammals, however, paternal effects are plausible in species with paternal care. Maternal effects on offspring behavior have been suggested to be lifelong, they have either a genetic or environmental basis and summarize the cumulative influence of many different proximate maternal effects, include pathways such as provisioning rates, milk production, in-utero hormone transfer, and epigenetics ⁹. Statistically, maternal effects account for similarities in dietary niche among offspring of the same mother (fitted as a random intercept for mother identity) ¹², ¹³, but not for the similarity of dietary niche between mother and offspring. The latter would be an example where the maternal trait affects the offspring’s trait, which statistically can be clearly differentiated from other maternal effects ¹², ¹³. Similarities between the dietary phenotypes of mothers and their offspring indicate social learning of resource preference or competence to secure a resource by the offspring from the mother during early ontogeny ¹⁷, ¹⁸, ¹⁹, ²⁰, ²¹. Social learning is therefore an additional pathway by which individual variation can be maintained. It is reasonable to assume that the effects of social learning during rearing will weaken later in life through individual experiential learning ²².”

I don't know what the recommendation is for the "brief material and methods" at the end of the introduction, but I think it's too detailed and I get lost between when you give your aims and results. You give the detailed methods at the end of the manuscript, so I won't spend too much time on it.

We have shortened this section although we deem it important to understand the results section. We think, given the journal format, that a reader needs to be able to understand the results without reading the detailed methods section. In response to another reviewers comment we also have referenced the different supplementary materials here to point readers to additional analyses. The objective and aim are repeatedly stated in the introduction line 72 “However, how individual variation in dietary specialization emerges and is maintained within populations has to our knowledge not been quantified in the wild.”; line 81 “Here, we provide the first study to attribute individual variation in dietary niche to its sources.”; line 110 “We used a unique 30-year longitudinal dataset of 71 female Scandinavian brown bears (*Ursus arctos*) of known mothers with repeated annual isotopic estimates of trophic position to study, for the first time, the sources of individual dietary specialization in the wild.”

L58 add (a.k.a. individual specialization) or L60 add individual specialization (i.e. among-individual variation of niche).

Changed line 65 to:

“Ecological generalists, species with a wide ecological niche, also seem to exhibit more individual specialization (i.e. between-individual variation of niche) ² and are likely particularly well adapted to persist under shifts in resource availability or composition, enabling them to occupy larger distributional ranges than ecological specialists ³.”

L63 Are you talking about among or within individual variation?

Among individual – should now be clear by adding it to first part of the sentence?

L63-65 I think the flow in the first paragraph will be better if this sentence is moved right after the first one (L59).

Changed as suggested

L66-68 Does this apply to all types of individual specialization or just individual dietary specialization?

Added dietary to emphasize that we are specifically referring to dietary specialization.

L-68- among-individual variation or both among and within?

The drivers listed and the reference given operated on the among population approach, i.e. populations with higher versus lower predation levels, while we are focusing on drivers among individuals within one population (next sentence).

L-70 Four non-exclusive potential sources. Do you have any references to justify these four potential sources?

We have added the following references:

“Laskowski KL, Chang C-C, Sheehy K, Aguiñaga J. Consistent Individual Behavioral Variation: What Do We Know and Where Are We Going? Annual Review of Ecology, Evolution, and Systematics 53, 161-182 (2022).”, where quote: “Among-individual (co)variation in behavior is thought to be generated by (semi)permanent genetic, maternal, developmental, or long-lasting environmental effects.”

“White SJ, Wilson AJ. Evolutionary genetics of personality in the Trinidadian guppy I: maternal and additive genetic effects across ontogeny. Heredity 122, 1-14 (2019).”

L70-73 What I think is not clear here is that part of the maternal effect can be social learning and part of the maternal effect can also be (indirect) genetic inheritance. L82-83 (e.g. maternal learning) . Maternal learning might be one of the reasons of similarity between offspring and mothers.

Conceptually you are right, statistically though there is a difference.

A maternal effect is measured as a similarity of offspring of the same mother and NOT the similarity between offspring and mother, as it is fitted as a random intercept of mother identity. Theoretically, a mother with a high trophic position could produce several offspring (from different fathers) which all have a similar trophic position which is however distinct from their mother. Following “McAdam AG, Garant D, Wilson AJ. The effects of others’ genes: maternal and other indirect genetic effects. In: Quantitative Genetics in the Wild (eds Charmantier A, Garant D, Kruuk LEB). Oxford University Press (2014).” We are teasing these apart as we fit the maternal trait as fixed effect into the model, this way we can partition the maternal effect into trait similarity (we interpret it as social learning) and other maternal effects (including indirect genetic effects). We hope this is now clearer from the revised introduction.

L88-90 Multigenerational pedigree is not required if you do an experimental approach, so I would specify that this approach is required in wild populations.

Added “in the wild”

L113 Why imitation and not direct learning? and how maternal learning differs from maternal effect?

Removed imitation here as the direct pathway is unknown, line 138: “If mothers differ in dietary niches, these differences may be maintained in the population through offspring social learning from the mother (hereafter “social learning”).”

L118 Maternal effects (i.e. maternal genotype or maternal environment) should come when you first mention maternal effects.

Has been mentioned where maternal effects are first explained (l. 94: “Maternal effects on offspring behavior have been suggested to be lifelong, they have either a genetic or environmental basis and include pathways such as provisioning rates, milk production, in-utero hormone transfer, and epigenetics 2.”)

L120-121 I don't think that genetic and environmental differences are two exclusive explanations.

Rephrased to line 130: “As a potential pathway, milk quantity or quality 32 can vary among females due to genetic differences and/or differences in their home range quality...”

Results :

I agree that the maternal learning effect can fade over time, but so can the other maternal effects, such as the genetic and environmental maternal effects. Can you explain why you made the choice to account for time since separation for maternal learning but not for the other maternal effects (random effect part)? Ignoring time for the maternal effect could, in part, bias your estimate of heritability downward.

As for the first part of your question, accounting for time in the social learning effect was straight forward by including an interaction with time while for the random effects structure, especially for covariance matrices such as pedigree or environmental similarity, this is to our knowledge not possible. The only way forward to test whether heritability is more important in shaping dietary specialization in certain life stages would be to split the model into a juvenile and adult model, similar to the approach in: “White SJ, Wilson AJ. Evolutionary genetics of personality in the Trinidadian guppy I: maternal and additive genetic effects across ontogeny. Heredity 122, 1-14 (2019).”

While this would be very neat, we do not have the sample size to allow for a finer time-specific analysis, for example, if we considered the first 4 years after separation as developing and everything beyond as adult, our sample size per model would decrease to n = 112 (n individual = 42) and n = 101 (n individual = 29), respectively. Though White et al did find that maternal effects were strong in juveniles while they

were bound to zero for adults and all variance was attributed to additive genetic effects, the meta-analysis by Moore, Whiteman ² did not corroborate this effect: "In contrast, maternal effects did not weaken across the offspring life cycle for physiology or behaviour."

We have addressed your comment in the discussion line 277.

"Similar to the effect of social maternal learning fading over time, additive genetic and maternal effects could be life-stage specific with maternal effects being more influential in juveniles¹³, although evidence for this is mixed ¹⁰."

In addition, given that you have motherID and pedigree, I was wondering if there was any particular reason not to partition the maternal effect variance into genetic and environmental maternal effect.

We would not know how to further subdivide these effects, do you have a reference? We fitted maternal identity as random effect which arguable encompasses both. In addition we fitted a maternal trait as fixed effect (so called "hybrid model"). From the literature (e.g., McAdam AG, Garant D, Wilson AJ. The effects of others' genes: maternal and other indirect genetic effects.; White SJ, Wilson AJ. Evolutionary genetics of personality in the Trinidadian guppy I: maternal and additive genetic effects across ontogeny. *Heredity* 122, 1-14 (2019)) this is the state-of-the-art way of quantifying maternal effects.

I did not understand why the time x maternal trophic position was considered as maternal learning? You show in L119-137 of the supplementary material that the distance between offspring and mother increases with time. So your interaction time by maternal position effect might just be a spatial effect due to dispersal and not due to learning fading over time.

Under your comment, environmental similarity and with-it resource availability changes over time with increasing distance between mothers and daughters. While we cannot fully rule out that such an effect exists we deem it unlikely for the reasons outlined in the supplementary material 5, where we show that spatial autocorrelation of habitat seizes after 10km. Indeed, female dispersal in our population follows a spectrum (supplementary 5, median dispersal distance = 8km) with few dispersers but most daughters are philopatric and settle close to their mother's home range with on average 40% home range overlap (Hansen JE, Hertel AG, Frank SC, Kindberg J, Zedrosser A. Social environment shapes female settlement decisions in a solitary carnivore. *Behav Ecol* 33, 137-146 (2021)). As outlined in the supplementary material, given the Swedish landscape and highly mobile moose as primary driver of trophic position we do not believe that changes in resource availability during "dispersal" can explain increasing dissimilarity between mothers and daughters.

L162- You said 72 females in the introduction.

Indeed, 71 is correct, thank you for pointing this out.

L226-229 The legend labels need to be more precise: environment -> environmental similarity to be consistent with the text legend; permanent individual -> permanent environment (to be more consistent with quantitative genetic literature?). Heritability is a ratio and not the variance components of interest, so something like "direct additive genetic" would be better. Please make sure this figure is color blind friendly (same for figure 3).

Thank you for your comment, we have changed environment to environmental similarity. We have kept heritability, because we are indeed presenting ratios and not raw variance components in this Figure, We have changed the colors to a color-blind friendly palette. Though we are aware of terminology in quant. genetics, we decided to keep "permanent between-individual" instead of "permanent environment", which is more in line with terminology in behavioral ecology. We have now used the colorBlind palette from the scale package for figure 2. Color schemes in Figure 1B and Figure 3A and B are now matching the color scheme in Figure 2.

Discussion

The authors mainly discuss the results of maternal learning, which gives the feeling that they only studied this effect, which is not the case. There are other components that are interesting and have been poorly studied in the literature.

Thank you for your comment, indeed we are concentrating on the maternal learning effect as it is the most novel aspect to be studied in the field of dietary specialization. We have added other aspects to the discussion, line 275:

"Additive genetic effects on the other hand were insignificant providing no evidence for heritability of dietary specialization in this population. Similar to the effect of social maternal learning fading over time, additive genetic and maternal effects could be life-stage specific with maternal effects being more influential in juveniles¹³, although evidence for this is mixed ¹⁰. We were not able to quantify life-stage specific heritability and maternal effects due to sample size limitations."

L257 The 8% permanent effect can also be due to a cohort effect and many other things.

We deem it less likely that a cohort affect would cause lifelong between-individual variation in trophic position but we did attempt to explain the potential cause and consequences of this effect more thoroughly. The sentence now reads, line 288:

"In addition, variation linked to between-individual effects (in our study 8 %) could be associated to either uncontrolled variation in resource availability in the environment (i.e., ecological opportunity 4, 35)

or individual differences in resource preference. The latter could, for example, be caused by differential individual learning and demonstrates the potential for behavioral innovation in this population. Ultimately, between-individual variation in dietary specialization allows populations to adapt to changes in resource availability, such as, new invasive prey or declines in food items due to climate change.”

L262 Sometimes you have distinguished social and maternal learning, sometimes you have used them as synonyms.

That is true and we have revised the language to “social learning” throughout the manuscript as this has created some confusion.

L289- But if there is social learning, do you think that removing one problematic individual from the population will be enough?

In bears and other solitary species, yes, as social learning is limited to learning from the mother, in more gregarious species such as wolves, targeting a single individual would be difficult. This is discussed in the cited reference (Swan GJF, Redpath SM, Bearhop S, McDonald RA. Ecology of Problem Individuals and the Efficacy of Selective Wildlife Management. Trends Ecol Evol 32, 518-530 (2017)).

Methods

I think this section also needs more guidance for the reader. First, it is not clear on what number of individuals you did the quantitative genetic analysis. Then I think it would be nice to have the model equation to better understand all the effects that were tested. You have a lot of different variance components and it is not clear what all your components together represent. For example: permanent environment is between individual differences not explained by additive genetic, maternal effect, habitat similarity, etc. For the maternal learning vs. maternal effect variance components, you could perhaps specify that maternal effect it is the rest of the maternal effect that is not learning. Finally, it might be good to specify what percentage of females disperse and at what distance, so that we can better understand the biological conditions under which we can disentangle environmental similarity from other components.

The sample size for the analysis is provided at the beginning of the results –213 measurements from 71 individuals. We have annotated the text more clearly with mathematical formula, please see our revised Statistical analysis section from line 497:

“We applied a two-step modelling approach. First, we fitted a basic linear mixed-effects model to estimate individual dietary specialization as permanent between-individual variation (VI) in trophic position. For this we used repeated measures of the same individual and fitted individual random intercepts. We accounted for a nonlinear effect of age (second-order polynomial, scaled by the standard deviation). We extracted the variance in fitted values (VAge; variance explained by age), permanent between-individual (VI), and residual within-individual variance (VR) and estimated each component’s proportional contribution to the total phenotypic variance ($VP = VA_{Age} + VI + VR$) through variance standardization 76.

Second, we used a spatially explicit Bayesian hierarchical model (i.e. ‘animal model’) 5, 40, 41 to partition permanent between-individual variance (VI) in trophic position into its sources; the fixed effects age (VAge) and social learning (VSL), and the variance components permanent between-individual variance (VI), environmental similarity (VE), additive genetic variance (VA), maternal effects (VM), and residual within-individual variance (VR). We followed the “hybrid” strategy suggested by McAdam, Garant 14 and tested for social learning of trophic position from the mother (“social learning”) by incorporating maternal trophic position as a fixed effect into the model. Thus, VM pools the remaining phenotypic variation of offspring trophic position that cannot be explained by maternal trophic position. We accounted for a nonlinear effect of time since separation of mother and daughter (scaled by the standard deviation), and for an interaction between maternal trophic position and time since separation to account for a decrease of the social learning effect over time. Age and time since separation were perfectly correlated: Pearson correlation coefficient > 0.99.

We calculated the total variance explained by the model ($VP = VA_{Age} + VSL + VI + VE + VA + VM + VR$) and calculated the proportion of the total variance explained by each model component. For the fixed effects, we partitioned the variance explained into social learning over time (i.e. maternal trophic position and its interaction with time since separation) and age (i.e. the main effect of time since separation), respectively, by calculating the independent contribution of each component to the total variance explained by the fixed effects, following the approach by Stoffel, Nakagawa 77 adapted to a Bayesian framework (see code under 78). For all parameters, we report the median and mean as measures of centrality and 89% credible intervals, calculated as equal tail intervals, as measure of uncertainty 79, 80. We deemed explained variance proportions as inconclusive when the lower credible interval limit was < 0.001 (i.e., < 0.1%) 81.

We have specified that maternal effects are the remaining variance not incorporating variance explained by maternal trophic position, line 513 (see above).

We have added information on dispersal in line 442:

“The median dispersal distance of daughters, i.e., the distance between natal and settlement home range centroids in this study was 8.56 km (range 1.4 – 28.8 km).”

We have added to the supplementary material S6 and show that spatial autocorrelation of home range habitat composition ceases after 10 - 15km, yet with a median dispersal distance of 8.56km we would not

expect that dispersing females necessarily inhabit very different habitats than philopatric females. We have referenced this in line 468:

“Spatial autocorrelation of home range habitat composition seized after 10 – 15 km (Fig S6).

L399- Please provide quick pedigree summary statistics (such as the number of generations).

We have added that out pedigree spanned 6 generations.

L433 It would be good to know the average home range size of the bears so that we can find out that 25 m spatial resolution is good.

This is a good point, also regarding your comment above concerning dispersal over time. We have added the following sentence line 453:

“Median lifetime home range size was 256 km², which is comparable to a circle with an 18 km diameter.”

I would say that a spatial resolution of 25 m is sufficiently detailed.

L458-459-472 You should add that you calculated the relative importance of each variance component and the associated credible interval, since these are the values you provided in the text. It might also be nice to provide the estimated variance of the full model in the supplementary materials.

We have added a line stating that we calculate the proportion of the total variance explained by each component line 519:

“We calculated the total variance explained by the model (fixed, random, and residual) and calculated the proportion of the total variance explained by each model component.”

We have also added the full summary table (median +/- 89% ETI) for fixed and variance components in the supplementary material S11 and are pointing towards that table in line 204.

Supplementary

L78-79 : Why is it written "among individual variation explains all the variance in trophic position" when the R² is 45% ?

Indeed, that sounds confusing. We meant that age does not explain any variance and all explained variance is attributed to among-individual variation. We have changed the sentence (line 78).

“Age accounted for 22% ($R^2_{\text{marginalmale}} = 0.26 [0.15, 0.37]$) of the variation in trophic position in male bears while it accounted for no variance of the trophic position in female bears ($R^2_{\text{marginalfemale}} = 0.01 [0, 0.02]$). Therefore, for females, all explained variance in trophic position was attributed to among individual differences ($R^2_{\text{conditionalfemale}} = 0.45 [0.31, 0.58]$).”

L148-147 : I don't understand why the spatial distance approach is better for this particular point (since it gives no information about ant or moose density). Individuals close in space may have similar phenotypes, but you will not know if this is due to similar moose/ant densities.

That is a good point, and particularly true for ant density which is tightly linked to habitat types, i.e. environmental composition. For moose, a very mobile species I would expect that population density changes more gradually over larger distances. We have rephrased this part, it now reads line 171:

“While environmental similarity incorporates aspects of habitat composition in the environment, it does not account for unmapped aspects such as resource density. Where resources change in a continuous fashion across the study area for reasons other than habitat composition, environmental similarity may not accurately reflect resources availability. For example, ant or berry availability will be tightly linked to the availability of suitable habitats, while moose density (a highly mobile species) will likely change over larger spatial scales and could be better captured using pairwise spatial distance among bear home ranges rather than environmental similarity.”

L156-157 Similar trophic positions due to social learning or shared genes.

Added

L170-171 It will be worthwhile to provide the autocorrelation coefficients by distance class so that we can understand at what spatial scale your habitat composition is autocorrelated and whether it's less than the dispersal distance.

Thank you for your comment.

We have provided an additional figure to the supplementary material calculating correlograms of different forest successional stages and the Simpsons Diversity index. From the Figure we can see that home ranges which's centroids were at very close spatial distance (10 - 15km) were spatially autocorrelated, likely because home ranges overlapped. At longer spatial distances, correlation coefficients dropped. The median female dispersal distance in our population is 8km.

L184 Why 98% and not 89% as in the main text?
This was a typo, thank you for pointing this out.

Reviewer #3 (Remarks to the Author):

I appreciate the attempts of the authors to pull together this information across such a long-term dataset. In revisions, it would be helpful to focus on several specific points to assist the reader in determining the validity of your work:

1. Lines 123-159 in the Introduction reads more like Methods than Introduction material. The authors even include references to Fig. 1A and Supplemental material before any results are included. I would suggest the authors include readily defined hypothesis, objectives, or predictions then use those for organizing the methods and results.

Our objective is to study the sources of dietary specialization and has been voiced this way in lines 69, 79, and 107. We introduce plausible pathways and cite previous studies how each of these sources could determine individual variation in diet. At the heart of the objective is a complex animal model which we deem important to understand the results section. Given that in this journals' format the method section is presented at the end of the paper we kept this short outline of the methods at the end of the introduction. We will leave it to the journal editor to determine whether this section should be removed. Another reviewer indeed suggested to include more reference to the supplementary material and we therefore kept the references in place for now.

2. The amount of details on stable nitrogen isotopes timing in hair (Lines 126-131) appears to be Methods that are placed into the objectives or it could be introductory material that might be better in a separate paragraph before your objectives. It appears that the majority of your analysis is based on accumulation of diet in hair so perhaps including these details in a separate paragraph, and adding more detail on how it has been done in bear or other species, would set up your final paragraph better than how it is currently constructed using Fig. 1?

We have kept Figure 1A as we deem it a nice visualization for a complex time line but we have revised that section as a separate paragraph, see lines 141:

“Individual trophic position is one metric to assess individual specialization along a continuum from a more plant-based to a more meat- or insect-based diet. Trophic position is calculated as a ratio of stable-nitrogen isotopes ($\delta^{15}\text{N}$) deposited into growing tissue and reflects cumulative diet intake during tissue growth. Individuals with higher trophic positions are specialized on more protein rich diets, relative to individuals with lower trophic positions which are increasingly more herbivorous. Trophic position therefore does not provide information on specific dietary items^{5, 6} or individual variation in niche breadth⁷ but rather quantifies the consumption of animal matter relative to other individuals in the

population. We here analyzed annual trophic positions from $\delta^{15}\text{N}$ in bear hair keratin⁸. $\delta^{15}\text{N}$ is deposited into bear hair with a delay of approximately one month (i.e., a growing hair in June reflects the diet intake in May⁹). Bear hair is annually renewed through molting in June, regrows over the summer and fall and stops growing during winter hibernation (Fig 1A,^{10,11}). Guard hair samples collected in spring therefore reflect annual estimates of the cumulative protein intake of individuals during the previous active season¹⁰.”

3. Based on 1 above, it is not clear in the Methods, which of these methods were needed for each objective because each objective is loosely presented.

In the introduction line 155 we introduce our two-step approach of first estimating individual dietary specialization and second estimating its sources.

“Using repeated samples of known mother-daughter pairs, we first estimated the extent of dietary specialization as permanent individual variation and second fitted a spatially explicit Bayesian hierarchical model (i.e. ‘animal model’)^{12, 13, 14} to quantify its sources”

In the statistics section of the methods the estimation of the two respective models are clearly differentiated. We added a sub header to the methods reading “Sources of individual variation in trophic position” with subsections: “Environmental similarity, Genetic pedigree, Maternal identity, Maternal trophic position (i.e. social maternal learning)”.

4. Related to 3 above, if hair is from “known” mother-daughter pairs (Line 132), and the methods in the Pedigree section were from Frank et al. , then there is no need to include this in your methods. I would suggest you state that mother-daughter pairs were determined in Frank et al. and not include the paragraph Genetic Pedigree and parentage assignment in the Methods because that component was already completed prior to this effort.

We included this section because the pedigree was used to estimate additive genetic variance. While indeed, mother-daughter pairs were known from following marked families during rearing, fatherhood is cryptic in bears. Mother daughter pairs were therefore not determined using the pedigree but rather using long-term monitoring, while the pedigree was used to estimate genetic heritability and should therefore be kept in the methods. Following your comment we have reduced the level of detail in this section, it now reads, line 472:

“A genetic pedigree based on 16 microsatellite loci was available for the population including 1614 individual genotypes, spanning six generation¹⁵. All female offspring and mothers in this study were genotyped and included in the population’s genetic pedigree. We used Cervus 3.0¹⁶ for assignment of fathers and COLONY¹⁷ for creating putative unknown mother or father genotypes and sibship reconstruction (see for details¹⁵).”

We introduced a separate section for maternal identity following comment 3 above, line 478.

Fig. 1A: It is not clear to me the need or intent for this figure? It seems citations confirming the relationship of diet to hair and hair growth across bear and different species as suggested above would be more valuable than a figure of this nature. In addition, the text cites Jimbo et al. and Cattet et al. for Fig. 1A so was this created and presented in those publications? If so, it should not be included here. If it was created in this effort then it should be in Methods and a Result of this work and not in the Introduction.

Figure 1a is a schematic overview depicting the complex time schedule of events between hair growth and feeding, hair sampling and molting; given the ecology of the study species. This Figure is based on previous work of Jimbo and Cattet but was not presented there. In our work we use Figure 1a to aid the reader in the introduction to explain why we believe that our sample collection is valid to represent our study topic. We therefore kept the Figure as part of the introduction.

Some specific comments by line below with the above points in mind:

Methods

Lines 367-370: The paragraph states that natural foods, plural, were collected but only moose hair was collected. Why was only moose hair and not muscle collected considering bear likely are not eating a great deal of hair? Would the authors expect moose hair and muscle to be similar in nitrogen stable isotopes or were both measured to confirm? It is not clear to me that hair is a good measure or comparable measure of muscle or reflective of stable nitrogen isotopes that would be incorporated into bear hair.

Thank you for our thoughtful comment.

We indeed collected samples of other food resources and their stable isotopic signature is shown in Figure S1. We did not run mixing models but calculated trophic positions which reflect each bear’s nitrogen signature along a continuum. Indeed, we opportunistically sampled moose hair from bear kill sites. As bears consume these carcasses quickly and we visited sites with a delay of a few days (kills were located from clusters of GPS samples), muscle tissue was mostly consumed or degraded and a standardized sampling of known soft tissue was not possible. In fact though, bears eat all parts of the moose - soft tissue and considerable amounts of hair.

In the supplementary material (S3.1) of a recently published study by our team (Mikkelsen, Ashlee J., Keith A. Hobson, Agnieszka Sergiel, Anne G. Hertel, Nuria Selva, and Andreas Zedrosser. 2023. “Testing

Foraging Optimization Models in Brown Bears: Time for a Paradigm Shift in Nutritional Ecology?" Ecology e4228. <https://doi.org/10.1002/ecy.4228>), we calculate the expected $\delta^{15}\text{N}$ of moose meat from sampled moose hair using published estimates of isotopic discrimination between ungulate hair and muscle under equilibrium conditions. According to the literature on bison, elk, caribou and moose, the discrimination for $\delta^{15}\text{N}$ between hair and meat varied between 0.77‰ in elk to 1.0‰ in bison and we took the mean ratio across these published studies, which was 0.88‰. Therefore:

$$\delta^{15}\text{N}_{\text{alces.alces.hair}} = 0.88 * \delta^{15}\text{N}_{\text{alces.alces.meat}}$$

Importantly, given the formula how trophic position is calculated, adding the correction factor does not change our results and only adds an arithmetic correction. That is because the stable isotopic signature of moose is only used to “scale” the bear nitrogen signature but it does not affect the relative distribution of bear samples to each other.

$$\text{Bear trophic position} = (\delta^{15}\text{N}_{\text{Ursus arctos}} - \text{average}(\delta^{15}\text{N}_{\text{alces.alces.hair}})) / 3.4 + 2$$

$$\text{Bear trophic position.corrected} = (\delta^{15}\text{N}_{\text{Ursus arctos}} - (\text{average}(\delta^{15}\text{N}_{\text{alces.alces.hair}}) / 0.88)) / 3.4 + 2$$

The range of trophic positions in bear hair calculated from the original moose samples was 2.41 - 3.95, whereas the corrected range would be 2.34 - 3.88. We opted to not rerun the analyses with a correction on moose for the following reasons: First, we did not sample moose meat in our particular study and the proposed correction would be based on estimates ranging from 0.77‰ to 1.0‰ thereby in our opinion adding a source of uncertainty rather than confidently correcting for bias. Second, brown bears in our study system indeed do feed on both muscle and hair, see line 417:

“Bears consume most of a moose carcass, including meat, skin, and hair. Soft tissue samples of moose carcasses could not be obtained but according to the literature the ratio of $\delta^{15}\text{N}$ in ungulate hair to meat ranges between 0.77‰ – 1.0‰. (see S3.1 in 67). We consider trophic positions calculated from moose hair representative and a correction of the $\delta^{15}\text{N}$ moose hair signature would only add an arithmetic correction but not change the distribution of bear trophic positions.”

It appears that other food items were collected (Supplemental Figure S1) but only moose were referenced here in the methods?

We used moose as a baseline to calculate trophic position. We did not use mixing models to calculate individual dietary components and we therefore did not deem it important to list these in the main text. We have added the following sentence, line 416:

“We calculated the trophic position of each bear hair sample relative to the average $\delta^{15}\text{N}$ value of moose hair representing trophic level 2 (mean \pm sd = 1.8 \pm 1.26 ‰, n = 21, Fig S1).”

Line 406: Citation 61 is not complete so please include additional details.

Thank you we have updated the citation details.

Line 411: Is Supplementary analysis 3 different than Supplemental Figure S3? If not, terms should be consistent. **It is an extended analysis with a Figure.**

Figure S2 and S4 are not referenced in the manuscript.

Thank you, we have references these now in lines 490 and 169

Environmental Similarity: There are several clarifications needed with this data that I believe the authors should consider:

1. What is the sample size for these GPS and VHF collars that had >1000 GPS or VHF locations? It is difficult to

assess if every bear that hair collected during capture was analyzed (so sample size reflected in first line of the Results) or if the Environmental similarity was based on a subsample of this assuming GPS/VHF collars failed, bears died, etc., thus a smaller sample size than what was originally collared and had hair collected. Line 431-432 is not adequate to inform the reader of the sample size that this entire section and analysis was based. 2. 95% kernel density estimator – This estimator is lacking considerable information considering it was done with both autocorrelated GPS data and likely not autocorrelated data with VHF telemetry. The program used for the kernel density estimator should be included as well as the smoothing option would be a minimum amount of information needed here. It would also be helpful to understand the reason for this choice if it was to “extract habitat composition in each bear’s lifetime home range.” Do the authors believe this is the best choice of an estimator considering it has been documented to be one of these least representative estimators of home range across many species?

To comment 1 and 2:

We see that line 431 must have added confusion.

Our dataset pulled together information from very different sources (hair sampling, genetics, movement data), and the final sample size (n = 71) was the number of individuals for which we had information on all of these aspects. We have added information on the R package used to calculate home ranges. We rephrased the sentence to, line 448:

“Here, we accounted for environmental similarity by extracting habitat composition in each bear’s lifetime home range (n = 71). We constructed home ranges using a 95% kernel density estimator (kernelUD function) in the R package adehabitatHR¹⁸ using the reference bandwidth as smoothing parameter (option “href”). Bears were monitored for a minimum of 1000 GPS locations (n = 47) or were located via VHF on at least 25 days (n = 24).”

Using a traditional kernel density estimator (KDE) on autocorrelated data is known to underestimate the sizes of home ranges (e.g., Noonan et al 2019 “A comprehensive analysis of autocorrelation and bias in home range estimation”). However, since we used this method across all individuals, they are all subjected to this same bias in home range size estimation. Our collar take a GPS position every At the time it was computationally more straight forward to do KDE than to fit home ranges with, e.g., akde, given the number of individuals in our study. With akde one has to fit multiple movement models to each individual and select the best one using AIC, after which one can obtain the utilization distribution. This can take up to an hour per individual.

We would prefer to keep the analysis as is, however, we are prepared to revise our home range estimator, if the reviewer insists.

3. Extracting home range composition for the “median” year if multiple years were monitored needs additional details. This might be appropriate for bears monitored 1-3 years but if monitored 5-10 years I question the reliability of the Corine Landcover Maps specifically if they are static and collected/released annually or in 5 year intervals? The lack of details on this data source makes this difficult to evaluate.

This is a valid point. The way the model is specified controls for environmental similarity with a covariance matrix and to our knowledge, this effect cannot be variable over time. Therefore, similar to a genetic pedigree, environmental similarity among pairwise home ranges is treated as a constant. The Corine Landcover Map was static but manually updated by us based on annual data of forest clearcutting by the Swedish Forestry Agency (Skogsstyrelsen). We manually recategorized mature forest stands to clearcuts in years when they were harvested; 9 years after harvest we recategorized them as young forests and 20 years after harvest as mature stands.

The real question is – how much habitat change do long-lived individuals experience during their lifetime and is it enough to affect the environmental similarity matrix (i.e. clearcutting changing their home range composition so substantially that its similarity relative to other home ranges changes).

We have addressed this comment in the following way – for individuals with > 4 monitoring years, we extracted home range composition in each year we monitored their trophic position. We plotted these annual values (points) alongside the value for the median year used in our analysis (line) and alongside the data distribution across all individuals. In fact, our annually updated maps reveal extremely little annual change in home range composition proportional to the overall home range size. We are therefore confident that the environmental similarity matrix is robust.

We have added this Figure to the supplementary material S5:

“Alternatively, resource abundance within home ranges could also change over the lifetime of individuals if habitat composition changes. Though our Corine Landcover Map was static, we accounted for forestry activity by manually updating the map with newly emerging clearcuts, based on annual data of forest clearcutting by the Swedish Forestry Agency (Skogsstyrelsen). We manually recategorized mature forest stands to clearcuts in years when they were harvested; 9 years after harvest we recategorized them as young forests and 20 years after harvest as mature stands. As environmental similarity is fitted with a constant environmental similarity matrix, we here provide a validation that the proportion of forest cover (FigS5B) and disturbed forest (FigS5C) in a bear’s home range only shifted minorly over the lifetime of an individual. We cannot exclude the possibility that we missed to account for unregistered changes in habitat composition, in particular forest disturbance.”

And referenced it in the main text line 458:

“Annual changes in home range habitat composition were negligible (Fig S5).”

Reviewer #4 (Remarks to the Author):

The authors used a very impressive dataset with repeated individual annual estimates of bear trophic position (derived from stable isotope analysis of hairs) to study individual dietary specialization. They partitioned among-individual variance in trophic position to discern the relative contributions of social learning, genetic predisposition, environmental forcings, and maternal effects. They conclude that social learning and maternal effects are as important for individual dietary specialization as environment. This result is very interesting because individual dietary specialization is key for species’ resilience against environmental change. It is also very interesting given the paucity of such detailed analyses in the literature.

I thus agree with the authors that the question and the results are important, and had no issue with the analyses. I was also very impressed by the gigantic amount of work behind this study, and found enough details in the methods for the work to be reproduced (at least in principle).

Thank you for your overall positive evaluation of our manuscript, we hope we have addressed your comments appropriately.

I do have three main criticisms, however; 1) lack of discussion regarding the pertinence of trophic position to evaluate dietary specialization, 2) apparent mistakes in some of the numbers that are presented (sorry if I am wrong on that point), and 3) need to clarify the text. Below I detail each point or give specific examples.

TROPHIC POSITION VERSUS DIETARY SPECIALIZATION

A given trophic position could be obtained from various diets and from various levels of dietary specialization. For example, a protein-rich diet could be obtained from various proportions of predation versus scavenging, or from various proportions of moose versus ants. Similarly, a rather generalist bear (e.g., feeding on crowberry and some ants) could have a similar trophic position as a specialist bear feeding exclusively on bilberry. I don’t know enough the ecology of brown bears to evaluate whether these examples are realistic, but I am questioning the fact that trophic position is used to investigate dietary specialization without any discussion on this methodological issue.

Thank you for your thoughtful comment. Indeed, we agree with you that using individual trophic positions is an oversimplification of individual specialization given that stable isotopic signatures in bear hair are shaped by all food resources combined (and $\delta^{15}N$ in particular by predation on moose and ants,

and scavenging of carcasses). In our population, endpoints of the isotopic niche are shaped primarily by berries on the one end and moose/ants on the other end (see Mikkelsen 2023: <https://doi.org/10.1002/ecy.4228>). We have added text to the introduction what a trophic position can tell us in terms of individual specialization. Importantly, our approach does not account for individual variation in niche breadth, i.e., from dietary specialist to dietary generalist individuals, but uses repeated individual measures to determine if individuals are consistently different from each other in their trophic position; line 141:

“Individual trophic position is one metric to assess individual specialization along a continuum from a more plant-based to a more meat- or insect-based diet. Trophic position is calculated as a ratio of stable-nitrogen isotopes ($\delta^{15}\text{N}$) deposited into growing tissue and reflects cumulative diet intake during tissue growth. Individuals with higher trophic positions are specialized on more protein rich diets, relative to individuals with lower trophic positions which are increasingly more herbivorous. Trophic position therefore does not provide information on specific dietary items^{5, 6} or individual variation in niche breadth⁷ but rather quantifies the consumption of animal matter relative to other individuals in the population. We here analyzed annual trophic positions from $\delta^{15}\text{N}$ in bear hair keratin⁸. $\delta^{15}\text{N}$ is deposited into bear hair with a delay of approximately one month (i.e., a growing hair in June reflects the diet intake in May⁹). Bear hair is annually renewed through molting in June, regrows over the summer and fall and stops growing during winter hibernation (Fig 1A, ^{10, 11}). Guard hair samples collected in spring therefore reflect annual estimates of the cumulative protein intake of individuals during the previous active season¹⁰. ”

APPARENT MISTAKES IN SOME NUMBERS

72 bears (line 91) versus 71 bears (line 162)? There is perhaps a good reason for this difference, but I did not find it.

There was no good reason, this was a mistake, it should be 71 bears and we have updated this.

Environmental similarity accounted for 9% of phenotypic variation in trophic position (line 174) versus 12% (line 45)?

9% is correct – we double checked all numbers.

Line 174 – “9% [0.1 - 5%]”. I don’t see how the 89% credible interval can be [0.1 - 5%] given a median of 9%. **This should read 49%.**

I tried with no success to match the percentages given in Results with the width of the color bands in Fig. 2. **Given that we fitted a Bayesian model, our inference is based on distributions rather than point estimates. Therefore, the proportion of variance explained by a given model component is also a distribution from which we can calculate the median, mean, or mode describing its central tendency and various types of credible intervals for uncertainty. When power is high and posteriors are perfectly gaussian, the mean and median are near identical but when posterior distributions are wide or skewed these can vary. Reporting the median of a distribution is more conservative and robust than the mean, see Pick et al. 2023. However, given that we report the median of seven posterior distribution, it means that all medians together do not add up to 100% (while the means of the posterior distribution do add up to 100%). We have now also reported the mean and show the mean proportion variance in Figure 2.**

Pick et al (2023) Describing posterior distributions of variance components: Problems and the use of null distributions to aid interpretation, Methods in Ecology and Evolution, DOI: 10.1111/2041-210X.14200

NEED TO CLARIFY THE TEXT

Please always use the same term to name a given source of variation, except when defining that source of variation (in which case synonyms can be useful). For example, genetic predisposition (line 41), genetic heritability (line 46), genetic inheritance (line 71), and genetic relatedness (line 134) seem to be used interchangeably, which generates lots of confusion.

Thank you for your comment, in line with your and other reviewer comments we have revised the language in the manuscript. For this specific comment – we now strictly use the term “genetic heritability”.

Fig. 1B and Fig. S3C. Can’t you give a name and some units to the Y axis?

Unfortunately the density of the posterior distribution is unitless as it is a distribution.

The legend of Fig. 2 would be easier to read if variables were given in the same order as they appear on the figure, which ideally should be in the same order as they appear in Results. Please also better explain the two rows of the figure (Basic and Maternal learning) in the legend of the figure. In short, whereas the first paragraph of Results, and Fig. 2, give most of the Take Home message of the paper, they lack the clarity that they deserve. **We have revised the legend for Figure 2, it now reads:**

“Figure 2. Drivers of brown bear trophic position as proportions of explained variance (median of the posterior distribution). Brown bears showed individual dietary specialization which did not change with age (Basic model). This individual dietary specialization could be explained by social maternal learning, permanent individual effects, genetic heritability, environmental similarity, maternal effects, and residual within-individual components (Maternal learning model).”

Fig. 2 – can you indicate the percentages of variance directly on the figure, as this will help relate the figure to the text (for example, indicate 11% inside the blue box).

We have been going back and forth about this idea before first submission and have decided against it for the reasons outlined above (discrepancies between means and medians).

ADDITIONAL COMMENT

Line 257. You indicate that “variation solely linked to individual variation (in our study 8 %) demonstrates potential for behavioral innovation and the potential to adapt to changing conditions”. Can you better explain this proposition?

Thank you for your comment, we have expanded this section, it now reads, line 288:

“In addition, variation linked to between-individual variation (in our study 8 %) could be associated to either uncontrolled variation in resource availability in the environment (i.e., ecological opportunity^{7, 19}) or individual variation in resource preference. The latter could for example be caused by differential individual learning and demonstrates the potential for behavioral innovation in this population. Ultimately, between-individual variation in dietary specialization allows populations to adapt to changes in resource availability, such as, new invasive prey or declines in food items due to climate change.”

1. Mousseau TA, Fox CW. The adaptive significance of maternal effects. *Trends Ecol Evol* **13**, 403-407 (1998).
2. Moore MP, Whiteman HH, Martin RA. A mother’s legacy: the strength of maternal effects in animal populations. *Ecology Letters* **22**, 1620-1628 (2019).
3. Daniel I. Bolnick, *et al.* The Ecology of Individuals: Incidence and Implications of Individual Specialization. *The American Naturalist* **161**, 1-28 (2003).
4. Dochtermann NA, Schwab T, Anderson Berdal M, Dalos J, Royauté R. The Heritability of Behavior: A Meta-analysis. *Journal of Heredity* **110**, 403-410 (2019).
5. Bolnick DI, Svanbäck R, Araújo MS, Persson L. Comparative support for the niche variation hypothesis that more generalized populations also are more heterogeneous. *Proc Natl Acad Sci U S A* **104**, 10075-10079 (2007).
6. Estes JA, Riedman ML, Staedler MM, Tinker MT, Lyon BE. Individual variation in prey selection by sea otters: patterns, causes and implications. *Journal of Animal Ecology* **72**, 144-155 (2003).
7. Balme GA, le Roex N, Rogan MS, Hunter LTB. Ecological opportunity drives individual dietary specialization in leopards. *Journal of Animal Ecology* **89**, 589-600 (2020).
8. Deniro MJ, Epstein S. Influence of diet on the distribution of nitrogen isotopes in animals. *Geochimica et Cosmochimica Acta* **45**, 341-351 (1981).
9. Rode KD, *et al.* Isotopic Incorporation and the Effects of Fasting and Dietary Lipid Content on Isotopic Discrimination in Large Carnivorous Mammals. *Physiol Biochem Zool* **89**, 182-197 (2016).

10. Jimbo M, *et al.* Hair Growth in Brown Bears and Its Application to Ecological Studies on Wild Bears. *Mammal Study* **45**, 337-345, 339 (2020).
11. Cattet M, *et al.* Can concentrations of steroid hormones in brown bear hair reveal age class? *Conserv Physiol* **6**, (2018).
12. Thomson CE, Winney IS, Salles OC, Pujol B. A guide to using a multiple-matrix animal model to disentangle genetic and nongenetic causes of phenotypic variance. *PLoS One* **13**, e0197720 (2018).
13. Gervais L, *et al.* Quantifying heritability and estimating evolutionary potential in the wild when individuals that share genes also share environments. *Journal of Animal Ecology* **91**, 1239-1250 (2022).
14. Wilson AJ, *et al.* An ecologist's guide to the animal model. *Journal of Animal Ecology* **79**, 13-26 (2010).
15. Frank SC, *et al.* Harvest is associated with the disruption of social and fine-scale genetic structure among matriline of a solitary large carnivore. *Evolutionary Applications* **14**, 1023-1035 (2021).
16. Kalinowski ST, Taper ML, Marshall TC. Revising how the computer program cervus accommodates genotyping error increases success in paternity assignment. *Molecular Ecology* **16**, 1099-1106 (2007).
17. Jones OR, Wang J. COLONY: a program for parentage and sibship inference from multilocus genotype data. *Molecular Ecology Resources* **10**, 551-555 (2010).
18. Calenge C. The package "adehabitat" for the R software: a tool for the analysis of space and habitat use by animals. *Ecological modelling* **197**, 516-519 (2006).
19. Araújo MS, Bolnick DI, Layman CA. The ecological causes of individual specialisation. *Ecology Letters* **14**, 948-958 (2011).

REVIEWER COMMENTS

Reviewer #1 (Remarks to the Author):

The revised manuscript flows well and reads well. Points were clarified well. I have a few extremely minor comments/suggestions that I hope will further improve the manuscript.

L209-L211: I found this statement a bit vague. "siblings occupying consistently lower or higher" - does this mean that sibling A was always higher than the daughter and sibling B always lower (for example)? How did this translate to different individuals?

Rephrased to: "Additionally, daughters of the same mother (i.e., full- and half-siblings) occupied similar dietary niches with consistently lower or higher trophic positions (Fig 3B)."

L275: I suggest a slight rephrase: "were not significant, " rather than "were insignificant".

Changed

L288: I suggest "associated with" rather than "associated to".

Changed

L403: Can you please clarify that hair samples includes bear samples and moose samples?

Changed

L472 and 475: These should be plural: generations and reconstructions.

Changed

Reviewer #1 (Remarks on code availability):

I accessed the data, but did not run the code. The provided PDF under "learning of carnivory" demonstrated how the code was used. I did not feel that I could review the statistics in detail as I am not that familiar with some of the analyses. However, I think that the methods are reproducible and could be run using the information provided.

Reviewer #2 (Remarks to the Author):

I was also a reviewer of a previous version and I appreciate the effort made by the authors to respond to the four reviewers. The authors are running complex quantitative genetic models, attempting to take into account various effects, which is commendable. However, I still have concerns about the 'social learning' effect in this complex model.

I am concerned about the maternal trophic position fixed effect (also known as the social learning effect) and its interpretation. I agree that the hybrid approach you are using proposes to fit the maternal traits 'x' as a fixed effect, but most of the time, this is not done with the exact same trait between offspring and mother. Correct me if I'm wrong, but the model appears to be: Cub trophic position ~ maternal trophic position * time since separation + age + (1|ID) + (1|maternal ID) + (1|environmental similarity). Here, because the maternal and offspring phenotypic traits are the same, your 'social learning' effect comprises both additive genetic and maternal effects. In other words, you are investigating the genetic basis of trophic position while using maternal trophic position as a fixed effect, which itself has a genetic basis and is highly genetically correlated with offspring trophic position because it is the same trait.

The consequence of this is that you are 1) underestimating your additive genetic variance and thus heritability, and 2) discussing how social learning is important while the estimated percentage might be highly inflated. One way to quickly assess if this is a significant issue would be to run the same model without the 'social learning effect' and observe how it affects your variance components. However, please refer to Pick et al., 2019 (DOI: 10.1002/evl3.125) where they encountered a similar issue and applied additional equations to correct the problem described above.

Thank you for your feedback – I appreciate to have a second set of eyes on our quite complex model structure. Indeed, the model is almost set up as you describe above. We did however include the additive genetic effect and did not fit a separate effect for age. Age and time since separation are perfectly colinear with a lag of one year, i.e. almost all cubs separate at age 1. Therefore, the main effect of time since separation can be interpreted as aging, while its interaction with social learning (i.e., "maternal trophic position") is interpreted as fading of the impact of maternal social learning over time.

Cub trophic position ~ maternal trophic position * time since separation + (1|ID) + (1|maternal ID) + (1|environmental similarity) + (1|additive genetic).

We have rephrased the methods section to make this clearer:

"Since age and time since separation were perfectly correlated (Pearson correlation coefficient > 0.99) we accounted for age with a nonlinear effect of time since separation between mother and daughter (second order polynomial, scaled by the standard deviation). We further accounted for a decrease of the social learning effect over time by fitting an interaction between maternal trophic position and time since separation."

You propose that our “social learning effect” is problematic because we use the same trait (trophic position in daughters and mothers) which by definition should be genetically correlated as we expect that most traits in nature have non-zero genetic heritability: “maternal trophic position as a fixed effect, which itself has a genetic basis and is highly genetically correlated with offspring trophic position because it is the same trait.”

We have addressed your comment in the following way:

As you suggested, we refitted our model without the maternal trophic position effect and monitored whether variance was attributed to the additive genetic effect.

Cub trophic position ~ time since separation* + (1 | ID) + (1 | maternal ID) + (1 | environmental similarity) + (1 | additive genetic).

*synonymous to age

We have added this additional analysis to the supplementary material S8.

By not controlling for the effect of maternal trophic position on offspring trophic position, variance was attributed first and foremost to environmental similarity (increase from a median explained variance of 5% to 30%) and only marginally to additive genetic effects (increase from 2.5% to 5%).

You previously suggested to assess power and significance for the additive genetic effect. Given your comment above, we now considered that running the power analysis should be done on a completely reduced model without any fixed effects (i.e. especially without the social learning term). We have redone and update the power analysis in the supplementary material S12 with the following model:

Cub trophic position ~ 1 + (1 | ID) + (1 | additive genetic).

In this fully reduced model, variance is indeed evenly attributed to permanent between-individual (median 21%) and additive genetic effects (median 21%, 89% ETI 1% - 50%). None of the permuted runs explained more variance (range over 1000 models 0.3 – 15.7%) suggesting that in the absence of other confounding variables, our pedigree indeed has enough power to identify significant heritability.

In conclusion:

- We have enough power to detect heritability given our dataset (Var explained = 21% in a reduced model) but after adding alternative explanatory variables (maternal effects, social learning and environmental similarity) heritability only accounts for a median explained variance of 2.5%.
- We do not find evidence that heritability and social learning are confounded. When removing the term for social learning (i.e., maternal trophic position) from the model, unexplained variance is first and foremost attributed to environmental similarity. This means bears that inhabit similar environments have more similar diets – S8.
- Maternal effects and social learning are not confounded – S7 – variance is attributed to permanent between-individual effects
- Spatial distance is strongly related to maternal effects, social learning and additive genetic effects and overrides all of these variables (Var explained = 63%); S6 – female bears are philopatric leading to daughters settling close in space to mothers; in consequence sisters also settle close to each other, and ultimately related females settle close in space; our study population spans 170km.

Additionally, considering the interaction between the 'social learning' effect and time since separation, I find the interpretation of VSL complicated. This interaction implies that the percentage of VSL and heritability can change over time, a point which you later discuss in the manuscript. However, how did you accommodate the interaction between 'social learning' and time since separation when calculating the variance attributed to social learning? Was the percentage of VSL estimated for a median time since separation?

Indeed this is a very good point. We have adapted code from Stoffel & Nakagawa (2021) “partR2: partitioning R2 in generalized linear mixed models” to calculate the independent contribution of different fixed effect components to the overall variance explained by the fixed effects. In our case we fitted:

*Time since separation * Maternal trophic position*

We first calculated the total variance via the model matrix. Second, we calculated structure coefficients for 1. The main effect of time since separation and 2. the maternal trophic position and the interaction. The rationale here was that we did not find an effect of age in the simple model but since age and time are perfectly correlated we can harness the interaction to test whether the effect of maternal trophic position on offspring trophic position wanes over time.

Finally, to address your question regarding the partitioning of maternal genetic and permanent environmental effects, in the reference you cited (McAdam & colleagues), they stated: 'Given a pedigree structure with at least three generations' depth, the animal model can be further extended to partition VM into genetic (VMG) and environmental (VME) components, and to estimate the covariance between direct additive and maternal genetic

effects (COVAMG). In practical terms, this implies fitting an additional random effect with maternal identity linked to the pedigree, similar to what you do for individual identity to distinguish between additive genetic and permanent environmental effects. However, considering that your model already has to distinguish between many different components with a low sample size, I no longer believe it is advisable to add any additional effects to your model.

Thank you for pointing out and explaining how we can distinguish between maternal genetic and maternal environmental effects – I would agree with you that our sample size is already stretching the limits of the model but perhaps this will be a feasible approach for other traits with larger sample sizes or when revisiting this question in a few years' time with an additional generation and more genetic links. We have started to update and rework our pedigree which will add another generation and more links. I will keep your comment and explanation here in mind for future work. Thank you for your constructive feedback on our model.

Point-by-point comments

L49: Please also provide the value for heritability.

Added

L217-219: Partly confounded.

Revised the sentence and changed to “strongly related”:

“However, after excluding spatial distance, social learning and maternal effects, but not heritability, explained more variance in trophic position (S7), indicating that spatial proximity and maternal effects are strongly related in this female philopatric species, where settlement home ranges of daughters are often close in space to their mothers forming so called matrilineal assemblages.”

L287: Between-individual effects include both genetic and permanent environmental effects, so please specify 'permanent between-individual effects.

Added as suggested

L290: Are you referring to learning from sources other than the mother? I am not sure I understand what you mean by 'differential learning.

That sounded strange, agreed. Rephrased to “by individual learning later in life”

L292: The term 'adapt' suggests an evolutionary response; perhaps 'adjust' would be more appropriate here.

We would argue that both is possible – if between-individual variation in dietary specialization exists then a population might have the population to adapt (as in – evolutionarily) to a changing environment. On the other hand, if individuals are plastic, they can adjust to changes in food availability. We therefore kept “adapt”.

L322-324: Considering that a large percentage of phenotypic variation is attributed to environmental similarity, don't you think that changing the relocating bears of their habitats (translocation) could be a solution? If bears are plastic enough, their trophic position might be influenced by changes of the environment.

This is an interesting point that I would like to elaborate on – however, environmental similarity accounted for surprisingly little variance in foraging specialization (after changing the home range estimator only 5%, less than we expected and less than social or maternal effects).

Translocations of problem bears are regularly attempted, for example in North America. It is mind boggling how some translocated bears travel hundreds of kilometers back to their original home range. Another example – in Europe, bears were translocated from Romania and Slovenia to the Italian Alps as part of a reintroduction program. Unfortunately, some of these bears had been food conditioned (i.e., fed by tourists) previously and became problem bears when released in Italy. Infamously, a female translocated bear has in the meantime been re-captured and put into captivity while two of her offspring have been removed by wildlife management for their problematic behavior. These are observational examples but they demonstrate that unfortunately, bears are too smart for their own good and translocation often does not work well or impact their behavioral strategy.

L354: Does this mean you consider food resource similarity to be different from habitat similarity? How do you define habitat similarity?

In our study, habitat similarity is measured by similarity proportion of mature and disturbed forest habitat, and habitat diversity irrespective of food resources. However, one would expect that habitat and food resources are tightly linked (hence also our prediction that environmental similarity should explain a portion of individual specialization). Yet, the environment may not match exactly the food it provides when food is mobile. For example, prey population densities change over larger spatial scales than local habitat composition (driven also by spatial variation in human harvest, see S6). In the discussion of our manuscript, we suggest that when an animal selects a home range this may not only be governed by the habitat but rather by the food resources that it provides.

L488: How can you be certain that age is not confounded with among-individual effects?

Similar to your main comment that maternal trophic position might be confounded with V_A mass is likely heritable (given that size is heritable in our population) and therefore confounded with V_A and V_i . Age on the other hand is just a number and by definition not confounded with V_i or V_A . For example, had we fitted mass, bears of age 4 can be heavier or lighter due to a number of different reasons, while none of these apply to the simple age of the bear.

We rephrased the sentence to: “We concentrate on the relationship between age and trophic position as, unlike mass or size, age is not confounded with between-individual effects (i.e., age itself cannot be heritable unlike mass or size).”

L500: Please specify the total number of observations for the animal model. Is it 213 ? because in the Table S1 you have much more observations (n=335).

It is indeed 213. We used the model in the supplement (n=335) to reconstruct maternal trophic positions. The main model could only use a subset of the full data frame as it relied on measures of the maternal phenotypic trait (maternal trophic position) to account for social learning.

„We applied a two-step modelling approach on our final dataset of 71 female offspring with 213 repeated annual measures of trophic position.”

L506: If you partition V_i into its sources, V_i can't be cited as one of the sources! From what I've read, I understand $V_i = V_{age} + V_{SL} + V_i + V_E + V_A + V_M + V_R$.

Hmm – that is an interesting point. What we mean is that in a first step we estimate the amount of between-individual variance from repeated measures data (the degree of individual specialization) and in a second step we assess whether individuals are different for a number of reasons, while indeed we still retain the V_i effect in the model to account for permanent “unexplained” between-individual variation.

Model1: $Y = V_{age} + V_i + V_R$

with $V_i \sim 50\%$ of the total phenotypic Variance

Model2: $Y = V_{age} + V_{SL} + V_i + V_E + V_A + V_M + V_R$

with $V_i \sim 9\%$ of the total phenotypic Variance

L516-517: If age and time since separation are perfectly correlated, why do you include both as fixed effects?

We are sorry, this was a misunderstanding. Time since separation and age are perfectly colinear, we therefore only include time since separation, however the main effect (how trophic position changes over time), we interpret as aging, while the interaction with maternal trophic position can be interpreted how the effect of social learning fades over time/age.

L522: Time since separation and age are the same effect? Why use two different names for it? I am confused.

We understand the confusion – while the effect is the same, the interpretation varies. Bears may change diet because they age or because the time since they initially socially learned their preference increases. We solve this by considering the main effect of Age/Time as “aging” and the main effect of maternalTrophicPosition in combination with the interaction as “social learning dynamic over time”. This effect produces Figure 3A

Age/Time + MaternalTrophicPosition + Age/Time:MaternalTrophicPosition

L529-530: With a Gaussian distribution family?

Correct, added

L335 of the supplementary: Please provide the total number of observations in your supplementary material; repeated measurements are known to increase statistical power. Also isn't it 71 individuals?

We have added the sample size and corrected the number of individuals, this should read 71 individuals with 213 repeated samples.

Table S1: Why is it listed as 95% CI while you have 89% CI in your manuscript?

Thank you, for pointing out this mistake.

Supplementary L355-356 : You find a pvalue of 0,07 so if I understand well it means that 7 % of the permuted models had higher additive genetic variance than the model with observed dataset. Which means that in 93 % of case you have more genetic variance than by chance and thus genetic variance in trophic position exist, or maybe I have misunderstood ? (« if related individuals have similar dietary specialization, our observed pedigree should explain more variance than a permuted pedigree).

7% had higher additive genetic variance than the observed dataset and 93% had lower additive genetic variance than the observed dataset. The observed pedigree in this case could not identify statistically significantly more additive genetic variance than by chance (although I agree with you – this is marginal and goes back to the interpretation of p-values which is debatable).

But you conclude that you did not find any evidence of heritability because you have a CI overlapping 0,1 % while here your permutation results seems to indicate that you do have an genetic variance (and heritability) but that you are not able to quantify it accurately. Also what does a p-value of 0,07 mean in term of power ? In Pick et al 2023 they defined power as « the proportion of 'observed' datasets in which the p-value was below a nominal threshold of 0.05. » So I don't see how it reflect power in your specific case.

We used the suggestion of Pick et al. 2023 section 2.3 at the very end “We calculated a p-value for each ‘observed’ dataset, as the proportion of estimates in the null distribution that were higher than the estimate from that ‘observed’ data.” In our case:

$$\text{sum}(V_{i,\text{observed}} > V_{i,\text{null}_{1:1000}}) / 1000$$

If 7 (out of 1000) permuted datasets have a higher V_i than $V_{i,\text{observed}}$ this leads to $p = 0.007$.

Another way to deal with « statistical power » is to use the head size for which you previously found moderate heritability (10.1111/eva.12786, $h^2=0,24$ [0,06-0,38]) and subsample your dataset to calculate new heritability with 71 individuals and XXX observations. So if you don't find any heritability or really low heritability you can be sure that your model struggle to estimate genetic variance and that it doesn't mean there is no heritability (but also see my comment about your maternal fixed effect that might under estimates your V_a).

I see your point that our supplementary analysis may mix concepts of statistical significance and power. We did struggle to think of a convincing way to demonstrate power. However, we have changed this analysis in relation to your main comment to this revision: we have dropped all fixed effects from this model, only accounting for additive genetic and permanent-between individual effects (see response to main comment). All permuted datasets have lower additive genetic variance than our observed dataset suggesting that, in the absence of ALL other confounding effects (maternal effects, social learning and the environment), our dataset is big enough to identify significant and relevant additive genetic effects (21% [1% - 50%]).

Reviewer #3 (Remarks to the Author):

I agree with the other reviewers that this was an impressive dataset, analysis, and compilation of data, however, that is also where it is deficient in so many ways. The authors argue for individual diets of bears such that Lines 114-117: "Populations range from tracking food resource pulses, such as spawning fish, scavenging on ungulate carcasses or preying on ungulate calves, or feeding extensively on invertebrates, to populations using primarily fruiting plant-based diets." With this breadth of diet, we would expect a simple analysis to document this which would include a figure like Figure S1 of all bears in their dataset. Presumably, bear foraging on carcasses would be different from those foraging on fruiting plants are at least separation of bear pairs that exhibit different overall diet selection.

There is an impressive wealth of diet studies demonstrating the breadth of brown bear diets. The aim of our study was to determine the sources of this variation within populations.

Figure 1B shows the phenotypic variation of trophic position in the population. Given that Scandinavian brown bears are omnivores – all bears are expected to forage on all of these different foods, however the proportional composition of foods on the individual level might vary, which is what we argue here based on Figure 1B. We have added a second Panel to Figure S1 showing the raw stable isotopic signatures of bear hair and bear foods (scaled by their appropriate discrimination factors).

The authors dismissed my comments on moose hair and basically stated that moose hair is equal in isotopes to moose meat or tissue consumed by bear. I realize that bear consume hair with the carcass but bear are not digesting and assimilating hair as they are moose tissue. As stated by the authors, bear and moose hair reflects a diet over time whereas tissue is within weeks of consumption and assimilation in the bear. It is not clear how moose hair can be a proxy for diet of moose?

We agree that one of the greatest challenges facing successful application of stable isotope applications in dietary studies is the proper choice of discrimination factors. The task becomes more challenging with omnivores that access both plant and meat in their diets. This situation is the result of a general lack of controlled captive studies where diet and tissue isotope values can be manipulated over long periods and the general “quick fix” used by those who mechanically apply the +3.4‰ for $\Delta^{15}\text{N}$ regardless. “Solutions” to this dilemma have emerged in the form of R packages like SIDER but they are only as good as the metadata they are based on.

In our case, we were forced to use moose hair $\delta^{15}\text{N}$ as a proxy for moose in bear diets because moose meat (presumably representing the bulk of ingested tissue by bears) was unavailable. This was the approach used by Urton et al. (2005) in their boreal foodweb reconstruction involving wolves following suitable arithmetic correction to convert hair to meat and then meat to consumer tissue. However, most importantly, the recent review paper by Stephens et al (2023) really does represent the most current and in-depth analysis of this question. Importantly, they report in Supplementary Material that $\Delta^{15}\text{N}$ for a herbivore in a C3 system (ours) for hair is 4.0‰ and for muscle is 3.9‰. For vertebrate omnivores in a C3 system, $\Delta^{15}\text{N}$ for hair is 3.2‰ and for muscle is 3.2‰. This indicates that for $\delta^{15}\text{N}$ (the isotope we used for trophic level determination), it does not matter if we use bear hair or bear muscle for dietary reconstructions and for moose, there is no evidence for an isotopic difference between hair and meat. So, current literature suggests that our trophic analysis is relatively unchanged by variance in discrimination factors (again for $\delta^{15}\text{N}$ and NOT for $\delta^{13}\text{C}$) and that our conclusions stand.

Stephens, R.B., O. N. Shipley, R. J. Moll. 2023. Meta-analysis and critical review of trophic discrimination factors ($\Delta^{13}\text{C}$ and $\Delta^{15}\text{N}$): importance of tissue, trophic level, and diet source. *Functional Ecology*
Urton, E.J.M., and K.A. Hobson. 2005. Intrapopulation variation in gray wolf isotope ($\delta^{15}\text{N}$ and $\delta^{13}\text{C}$) profiles: implications for the ecology of individuals. *Oecologia*14:317-326.

Also, using KDE home ranges with reference bandwidth smoothing likely severely over-estimates the size of the area used by each bear. Even if we overlook the issue of GPS versus VHF telemetry in the sample size (Bears were monitored for a minimum of 1000 GPS locations ($n = 47$) or were located via VHF on at least 25 days ($n = 24$) so comparing home ranges of 1,000 to 25 locations as being equal, but centroids to core home ranges for home ranges that are considerably larger than what the bear uses makes this analyses difficult to interpret and justify. The authors do not provide this raw data, shapes of the home ranges used to determine overlap/distance between core areas, or how using 1000 versus 25 locations depending on bear would influence their results. Even in their rebuttal to my comments on this, a sentence is cut off mid way.

Thank you for your comment. In fact, KDE is known to underestimate home range sizes, not overestimate. To address your concern we have now re-estimated all home ranges using autocorrelated kernel density estimator (AKDE) from the ctm package. We have extracted home range composition in these new home range polygons to assess the effect of environmental similarity and have extracted new home range centroids from which we calculated pairwise spatial distance (used in the supplementary material). Importantly, home range sizes using either estimator were strongly correlated with most home ranges having an estimated larger home range size while a few others were estimated at smaller range sizes (see Figure below). As accordingly centroids shifted, pairwise spatial distances tended to become longer for a few bears with significant changes in estimated home range size.

We have rerun all statistical models in the main text and supplement.

Updating the home range estimator did not significantly change our results – the variance explained by environmental similarity decreased from a median 9% to a median 5%, while the variance explained by spatial distance increased from a median 50% to a median 63%.

The authors also dismissed my comment that figure 1A is not needed because it just repeats text and provides no further understanding of bear molting timeframe. I also have no idea what value there is in Figure 1B but if it is from a separate dataset, I don't see why it is included.

Figure 1B represents the posterior distribution of each individual ($n = 71$ female brown bears, the one dataset used for all analyses in the main text) and demonstrates individual specialization in trophic position.

How does the annual changes in home range habitat composition being negligible (Lines 456-458) show up in Figure S5 that focuses on distances between natal range centroids?

We have grouped a) annual changes in lifetime habitat composition, and b) annual distances between natal and settlement home ranges because both are sources of uncertainty in the way we accounted for the environment (environmental similarity matrix or spatial distance matrix). We fully agree but also disclose that using lifetime home ranges and a static environment centered on the median year of observation oversimplifies temporal dynamics of a) the environment and b) the space used by a bear. Yet, to date we cannot account for temporal variation in covariance matrices.

Reviewer #3 (Remarks on code availability):

The code is mostly output only code. Nobody with a comparable dataset could replicated their research and we are not even able to replicated what they assessed with so much data missing. There is no raw data to actually perform any analyses in a reproducible way. The authors saved .rds files after running analyses on raw data thus we can only evaluate what the authors chose to provide. They included code for reproducing figures that are already included in the manuscript which seems to have limited value.

The clean and complete dataset, i.e., without errors or missing data, that was generated and analyzed during this study and code to reproduce all analyses (including the supplementary analyses) is provided under the Open Science Framework: <https://doi.org/10.17605/OSF.IO/68B9U>. This is in line with Nature Communications Data and Code sharing policies.

Providing raw data as rds files is common standard. This file format is advantageous over other data formats as it is more stable (files cannot be opened and cell contents changed, no problems with decimal delimiter). Our final dataset indeed includes a column of model derived data, i.e., maternal trophic position. Code to reproduce this column and to derive this final dataset is provided in the Supplementary Code S1-S3.

Given that we run Bayesian models, we saved model outputs to reproduce exact point estimates but all code to run models is provided. This is also standard practice.

As the code is annotated, someone that is interested to replicate the study with their own data should be able to adapt our code to their own data.

Thus, the above statement is incorrect.

Specifically, there are no bear telemetry locations to reproduce the home range estimates.

It is correct that we do not provide raw GPS data but instead a preprocessed data frame of spatial distances and environmental composition.

We further do not provide the data and code to create home ranges and extract spatial covariates, for two reasons:

a) The spatial layers are products that we bought ourselves and we do not have the right to distribute these.

b) We do not provide raw GPS & VHF data for our long-term study as we are currently not in the position to make this database entirely open access. All data are archived in the wireless remote animal monitoring server at SLU in Sweden and we would provide access to data and code upon reasonable request. As an example, we have now included the raw GPS data of one individual and a script how home ranges were estimated using ctm in the OSF code & data repository. We have also updated our Data and code availability statement to:

“Primary data and code to reproduce all analyses are provided in the OSF repository; <https://doi.org/10.17605/OSF.IO/68B9U> 78. Restrictions apply to the GPS data and code to reproduce home ranges; these can be obtained from the first author upon reasonable request.”

They included bear C13 and N15 for 71 bears but some females had 1-7 daughters and appear to be considered independent or separate samples.

The model in the main text analysis N15 for 71 female bears with 213 records which are daughters with known information on their mother’s trophic position (n=33 mothers). We analyze stable isotopes for all bears in the Supplementary material S2 & S3 and both code and raw data are provided in the repository. In fact, the cleaned dataset used in the main analysis is built in the code “Supplement S1_S3.rmd”, it is therefore reproducible how we derived the final dataset from the total dataset.

I understand if the other reviewers provide a more useful review for the journal but figured I would provide what I could.

Reviewer #4 (Remarks to the Author):

The questions I had raised during my initial review have been answered with satisfaction, the needs for clarification have been taken into account, and the mistakes I had pointed out have been corrected.

I am still impressed by the gigantic amount of work behind this study and have no doubt it represents a significant advance in the field. I congratulate the team for their research.

Thank you for your positive feedback

Point by point response:

We thank the reviewer for their time reading and commenting on our manuscript. After three rounds of review, the reviewer had four remaining general comments, which we answer below. As these comments were based on already two rounds of review, we have appended the relevant previous comments by the reviewer and our rebuttals at the bottom of this document.

Reviewer #3 (Remarks to the Author):

Reviewer comment 1. “There is no point to Figure 1A considering it is a diagram of the exact sentence (Lines 150-151) stating: Bear hair is annually renewed through molting in June, regrows over the summer and fall and stops growing during winter hibernation.”

Response:

As already outlined in **rebuttal 1 and 2** - Figure 1A is a schematic overview depicting the complex time schedule of events between hair growth and feeding, hair sampling and molting.

We specifically prepared and included this Figure to aid the readers understanding, while indeed we also explain the sequence of events in the manuscript text lines 149 – 153:

“Hair $\delta^{15}N$ represents a dietary integration of about a month (i.e., a growing hair in June reflects the diet intake since May³⁷). Bear hair is annually renewed through molting in June, regrows over the summer and fall and stops growing during winter hibernation (Fig 1A, ^{38, 39}). Guard hair samples collected in spring therefore reflect annual estimates of the cumulative protein intake of individuals during the previous active foraging season³⁸.”

37. Rode KD, et al. Isotopic Incorporation and the Effects of Fasting and Dietary Lipid Content on Isotopic Discrimination in Large Carnivorous Mammals. *Physiol Biochem Zool* 89, 182-197 (2016).

38. Jimbo M, et al. Hair Growth in Brown Bears and Its Application to Ecological Studies on Wild Bears. *Mammal Study* 45, 337-345, 339 (2020).

39. Cattet M, et al. Can concentrations of steroid hormones in brown bear hair reveal age class? *Conserv Physiol* 6, (2018).

The reviewer has repeatedly expressed their opinion that Figure 1A is not needed, yet, we the authors of the manuscript deem the inclusion of this Figure useful. We are prepared to remove Figure 1A from the manuscript for editorial reasons.

For reference: Figure 1 and legend, as included in the current version of the manuscript.

„Figure 1. A) Bear hair generally grows from June until October and stable-nitrogen isotopes ($\delta^{15}N$) reflects cumulative diet intake during the period of hair growth. The quiescent phase, when hair ceases growing, lasts through hibernation, followed by emergence from the winter den and molting in late May-early June. Hair samples were taken during bear captures in April - June and reflect the bears' diet in the previous year ...”

Reviewer comment 2. “Stable isotopes of moose hair being used when the authors acknowledge that a majority of diet is moose meat but moose meat was not available for stable isotope assessment. The authors then cite Stephens et al. (2023) that “arithmetic correction to convert hair to meat” was 4.0 per mil for hair and 3.9 per mil for meat so there is no difference between the two. My concern is that

correction factors aside, we simply don't know. What we do know, and what was confirmed by the authors, was that bear don't consume moose hair but they do consume moose meat.”

Response:

Most importantly, and as explained in **rebuttal 1**, for our analysis it is **irrelevant whether we use moose hair or moose meat for trophic position calculation**. We could have also used the $\delta^{15}\text{N}$ signature of any of the three berry species consumed by brown bears (bilberry, crowberry, and lingonberry) or even the raw $\delta^{15}\text{N}$ isotopic values of bear samples without any information on consumed resources and would have obtained the exact same results regarding the drivers of dietary specialization. This is because the trophic position (TP) of the consumer is calculated as the difference between the $\delta^{15}\text{N}$ signature of the consumer and the resource divided by a trophic discrimination factor (DF) plus the trophic position of the resource, i.e.:

$$\text{TP}_{\text{consumer}} = [(\delta^{15}\text{N}_{\text{consumer}} - \delta^{15}\text{N}_{\text{resource}}) / \text{DF}] + \text{TP}_{\text{resource}} \text{ (Post, 2002).}$$

From the formula above it is clear that **the conversion does not change the relative distribution of bear nitrogen estimates**, but only scales their absolute value relative to a baseline resource with known trophic position (e.g., a strict herbivore or consumed plants). Since the $\delta^{15}\text{N}$ signature of the resource is only used for arithmetic scaling, our results remain unchanged regardless of the exact $\delta^{15}\text{N}$ signature of the dietary item used to construct trophic positions (e.g., moose hair or meat). **Reviewer 3's comment therefore does not pertain to the conclusions drawn in our paper.**

Regardless, the reviewer is incorrect in their assessment stating that we have no knowledge on isotopic difference between hair and meat, as we have explained in **rebuttal 2**. There is published evidence that moose hair and moose meat have similar discrimination factors (Stephens et al. 2023, detailed in rebuttal 2). This indicates that for $\delta^{15}\text{N}$, there is **no evidence for an isotopic difference between hair and meat**. In addition, using either moose meat (with an approximated a correction factor taken from the literature) or moose hair for trophic position reconstruction in our system did not qualitatively change bear trophic position estimates and had no effect on its interpretation regarding a more herbivorous or more carnivorous diet (detailed in **rebuttal 1**).

In conclusion, reviewer 3's concern regarding differences in discrimination factors of moose hair and meat is not supported by the literature (see Stephens et al. 2023) and ultimately, any differences in discrimination factors would not affect our analysis or conclusions because in calculating consumer trophic position (i.e., bear) dietary isotopic signature (i.e., moose) is only used for arithmetic scaling in.

Post, D.M. (2002) Using stable isotopes to estimate trophic position: models, methods, and assumptions. *Ecology*, 83, 703-718. doi: [https://doi.org/10.1890/0012-9658\(2002\)083\[0703:USITET\]2.0.CO;2](https://doi.org/10.1890/0012-9658(2002)083[0703:USITET]2.0.CO;2)

Stephens, R.B., O. N. Shipley, R. J. Moll. (2023), Meta-analysis and critical review of trophic discrimination factors ($\Delta^{13}\text{C}$ and $\Delta^{15}\text{N}$): importance of tissue, trophic level, and diet source. *Functional Ecology*, <https://doi.org/10.1111/1365-2435.14403>

Reviewer comment 3. “Size of home range - The authors claim that: "In fact, KDE is known to underestimate home range sizes, not overestimate." The authors should be required to show supporting citations to this statement but they are completely wrong. The authors did investigate another KDE estimator and found the centroid distances shifted or distances increased for some bear, again supporting that KDEs to overestimate sizes of home range.. Trading out one KDE estimator (reference bandwidth smoothing) for another (AKDE) does not really address my comments.”

Response:

In our manuscript we estimate lifetime home ranges and extract habitat composition within these home ranges to examine whether differences in the environment (differences in home range habitat composition) explain differences in dietary specialization. Reviewer 3's main concern was that we originally used Kernel Density Estimator (KDE) with reference bandwidth smoothing to construct home ranges. The reviewer claimed that KDE is overestimating home range size.

We assume that the **reviewer's notion that KDE overestimates home range size** is based on older studies, prior to the invention of AKDE (the home range estimator that we are now using). For example (Downs and Horner, 2008) concluded: “*The KDE methods ... overestimated home ranges by about 40-50% even in the best cases. For convex, linear, perforated, and disjoint point patterns, KDE methods overestimated home-range sizes by 50-300%, depending on sample size and method of bandwidth selection.*” However, **more recent studies** in fact have shown the opposite, e.g. Fleming et al. (2015): “*...using realistically autocorrelated data in conventional KDEs results in grossly underestimated home ranges.*” (also, Fleming & Calabrese 2017, Noonan et al. 2019, as detailed in **rebuttal 1** where we also included the respective citation). Reviewer 3's opinion on KDE is therefore incorrect, as of the current state of the literature.

Regardless of the direction of bias, to address reviewer 3's comment we had already changed our home range estimator in **rebuttal 2**, based on the same reviewer comments, from KDE (Kernel Density Estimator) to AKDE (Autocorrelated Kernel Density Estimator), which accommodates the properties of our movement data better: (a)

GPS locations collected at regular intervals which are inherently autocorrelated, and (b) VHF locations which represent independent and identically distributed data. AKDE is currently the gold standard to estimate home ranges from tracking data and we are unaware of any better method (nor did the reviewer suggest a better or alternative method). Finally, as explained in **rebuttal 2**, the results of our analysis were largely unchanged by the choice of home range estimator.

Downs JA, Horner MW, 2008. Effects of Point Pattern Shape on Home-Range Estimates. *The Journal of Wildlife Management* 72:1813-1818. doi: <https://doi.org/10.2193/2007-454>.

Fleming, C.H., Fagan, W.F., Mueller, T., Olson, K.A., Leimgruber, P. and Calabrese, J.M. (2015), Rigorous home range estimation with movement data: a new autocorrelated kernel density estimator. *Ecology*, 96: 1182-1188. <https://doi.org/10.1890/14-2010.1>

Fleming, C.H. & Calabrese, J.M. (2017) A new kernel density estimator for accurate home-range and species-range area estimation. *Methods in Ecology and Evolution*, 8, 571-579. doi: <https://doi.org/10.1111/2041-210X.12673>.

Noonan, M.J., Tucker, M.A., Fleming, C.H., Akre, T.S., Alberts, S.C., Ali, A.H., Altmann, J., Antunes, P.C., Belant, J.L., Beyer, D., Blaum, N., Böhnning-Gaese, K., Cullen Jr., L., de Paula, R.C., Dekker, J., Drescher-Lehman, J., Farwig, N., Fichtel, C., Fischer, C., Ford, A.T., Goheen, J.R., Janssen, R., Jeltsch, F., Kauffman, M., Kappeler, P.M., Koch, F., LaPoint, S., Markham, A.C., Medici, E.P., Morato, R.G., Nathan, R., Oliveira-Santos, L.G.R., Olson, K.A., Patterson, B.D., Paviolo, A., Ramalho, E.E., Rösner, S., Schabo, D.G., Selva, N., Serghi, A., Xavier da Silva, M., Spiegel, O., Thompson, P., Ullmann, W., Zięba, F., Zwijacz-Kozica, T., Fagan, W.F., Mueller, T. & Calabrese, J.M. (2019) A comprehensive analysis of autocorrelation and bias in home range estimation. *Ecological Monographs*, 89, e01344. doi: <https://doi.org/10.1002/ecm.1344>.

Reviewer comment 4. “Code and data availability - the authors state in their rebuttal: "Providing raw data as rds files is common standard. This file format is advantageous over other data formats as it is more stable (files cannot be opened and cell contents changed, no problems with decimal delimiter). Our final dataset indeed includes a column of model derived data, i.e., maternal trophic position. Code to reproduce this column and to derive this final dataset is provided in the Supplementary Code S1-S3. ”

This code and most data was not previously provided, I have it downloaded because I looked into this in the last revision. Some data is now available as .rds only because we instructed that it be made available.”

And in addition

Reviewer #3 (Remarks on code availability):

“This code and most data was not previously provided, I have it downloaded because I looked into this in the last revision. Again, the authors dismissed my comments that it was provided and I simply missed it. Some data is now available as .rds only because we instructed that it be made available. HOWEVER, this is not data. This is only output of analysis with some raw data. We can't generate home range isopleths from their code (the authors acknowledge they can't share GPS/VHF locations of bear) but they also don't give us shapes or sizes of resulting polygons from their home range analysis. The data they provide is equivalent to providing results in Tables in the text or Supplements so provides no value towards reproducible science in any way.

In conclusion, the authors provide considerable data of results and to make figures which is not the purpose for open science. I understand that the authors may not be able to share actual bear locations, but I am not able to determine shapes or sizes of home range and not able to assess their habitat composition without raw polygons of home ranges either. The data and code to run models is provided as confirmed by the authors so the analysis is reproducible for that portion of the manuscript.”

Response:

We provide complete data and code to reproduce the **main and all supplementary analyses**, as confirmed by reviewer 3 and also a second code availability reviewer and all code to reproduce the analysis had already been available from the beginning.

After revisions we had updated all files in the code and data directory (<https://osf.io/68b9u/>) on 2024/04/29 (these updates are logged in page 1 -11 of the “Recent Activity” tab). However, all supplementary files were already accessible when the reviewer first reviewed the code directory: For example, the file

“Supplement_S1_S3_MaternalPaternalTrophicPosition.Rmd” had been uploaded on 2023/03/23 as shown on page 12 of the “Recent Activity” tab.

The only exemption are the raw GPS/VHF location data of bears which are used to create home ranges because these are sensitive data and we simply have no permission to publish them. We also do not provide environmental layers that we do not have the right to distribute (see **rebuttal 2**), but have provided the source. Because of these data limitations we have not uploaded the code to a) produce home ranges and home range centroids from raw GPS data, and b) extract environmental covariates within these home ranges.

On request by reviewer 3, we had added a reproducible example to create home ranges for one bear. Indeed, this code was not originally available and we had specifically prepared it during **rebuttal 2** to address the reviewer's comment.

We emphasize that to our knowledge this is within the data sharing policy of Nature Communications and have communicated this limitation within the Data & Code availability statement. We will share the raw GPS data and code upon reasonable request.

While we cannot provide the raw GPS data, we have now included a map in the supplementary material Fig S6A and S6B, showing home ranges, habitat composition, and home range centroids.

A
Female bear home ranges
(n = 71)

B
Female bear home range centroids
(n = 71)

Additional information:

Previous comments by reviewer 3 from the first and second round of review:

1. The purpose of Figure 1

Reviewer comments & author rebuttal 1:

Fig. 1A: It is not clear to me the need or intent for this figure? It seems citations confirming the relationship of diet to hair and hair growth across bear and different species as suggested above would be more valuable than a figure of this nature. In addition, the text cites Jimbo et al. and Cattet et al. for Fig. 1A so was this created and presented in those publications? If so, it should not be included here. If it was created in this effort then it should be in Methods and a Result of this work and not in the Introduction.

Figure 1a is a schematic overview depicting the complex time schedule of events between hair growth and feeding, hair sampling and molting; given the ecology of the study species. This Figure is based on previous work of Jimbo and Cattet but was not presented there. In our work we use Figure 1a to aid the reader in the introduction to explain why we believe that our sample collection is valid to represent our study topic. We therefore kept the Figure as part of the introduction.

Reviewer comments & author rebuttal 2:

The authors also dismissed my comment that figure 1A is not needed because it just repeats text and provides no further understanding of bear molting timeframe. I also have no idea what value there is in Figure 1B but if it is from a separate dataset, I don't see why it is included.

Figure 1B represents the posterior distribution of each individual (n = 71 female brown bears, the one dataset used for all analyses in the main text) and demonstrates individual specialization in trophic position.

2. On the matter of using moose hair stable isotopes for trophic position reconstruction

Reviewer comments & author rebuttal 1:

Lines 367-370: The paragraph states that natural foods, plural, were collected but only moose hair was collected. Why was only moose hair and not muscle collected considering bear likely are not eating a great deal of hair? Would the authors expect moose hair and muscle to be similar in nitrogen stable isotopes or were both measured to confirm? It is not clear to me that hair is a good measure or comparable measure of muscle or reflective of stable nitrogen isotopes that would be incorporated into bear hair.

Thank you for our thoughtful comment.

We indeed collected samples of other food resources and their stable isotopic signature is shown in Figure S1. We did not run mixing models but calculated trophic positions which reflect each bear's nitrogen signature along a continuum. Indeed, we opportunistically sampled moose hair from bear kill sites. As bears consume these carcasses quickly and we visited sites with a delay of a few days (kills were located from clusters of GPS samples), muscle tissue was mostly consumed or degraded and a standardized sampling of known soft tissue was not possible. In fact though, bears eat all parts of the moose - soft tissue and considerable amounts of hair.

In the supplementary material (S3.1) of a recently published study by our team (Mikkelsen, Ashlee J., Keith A. Hobson, Agnieszka Sergiel, Anne G. Hertel, Nuria Selva, and Andreas Zedrosser. 2023. "Testing Foraging Optimization Models in Brown Bears: Time for a Paradigm Shift in Nutritional Ecology?" Ecology e4228. <https://doi.org/10.1002/ecy.4228>), we calculate the expected $\delta^{15}\text{N}$ of moose meat from sampled moose hair using published estimates of isotopic discrimination between ungulate hair and muscle under equilibrium conditions. According to the literature on bison, elk, caribou and moose, the discrimination for $\delta^{15}\text{N}$ between hair and meat varied between 0.77‰ in elk to 1.0‰ in bison and we took the mean ratio across these published studies, which was 0.88‰. Therefore:

$$\delta^{15}\text{N}_{\text{alces.alces.hair}} = 0.88 * \delta^{15}\text{N}_{\text{alces.alces.meat}}$$

Importantly, given the formula how trophic position is calculated, adding the correction factor does not change our results and only adds an arithmetic correction. That is because the stable isotopic signature of moose is only used to "scale" the bear nitrogen signature but it does not affect the relative distribution of bear samples to each other.

$$\text{Bear trophic position} = (\delta^{15}\text{N}_{\text{Ursus arctos}} - \text{average}(\delta^{15}\text{N}_{\text{alces.alces.hair}})) / 3.4 + 2$$

Bear trophic position.corrected = ($\delta^{15}\text{N}_{\text{Ursus arctos}} - (\text{average}(\delta^{15}\text{N}_{\text{Alces alces.hair}}) / 0.88)) / 3.4 + 2$

The range of trophic positions in bear hair calculated from the original moose samples was 2.41 - 3.95, whereas the corrected range would be 2.34 - 3.88. We opted to not rerun the analyses with a correction on moose for the following reasons: First, we did not sample moose meat in our particular study and the proposed correction would be based on estimates ranging from 0.77‰ to 1.0‰ thereby in our opinion adding a source of uncertainty rather than confidently correcting for bias. Second, brown bears in our study system indeed do feed on both muscle and hair, see line 417:

“Bears consume most of a moose carcass, including meat, skin, and hair. Soft tissue samples of moose carcasses could not be obtained but according to the literature the ratio of $\delta^{15}\text{N}$ in ungulate hair to meat ranges between 0.77‰ – 1.0‰. (see S3.1 in 67). We consider trophic positions calculated from moose hair representative and a correction of the $\delta^{15}\text{N}$ moose hair signature would only add an arithmetic correction but not change the distribution of bear trophic positions.”

It appears that other food items were collected (Supplemental Figure S1) but only moose were referenced here in the methods?

We used moose as a baseline to calculate trophic position. We did not use mixing models to calculate individual dietary components and we therefore did not deem it important to list these in the main text. We have added the following sentence, line 416:

“We calculated the trophic position of each bear hair sample relative to the average $\delta^{15}\text{N}$ value of moose hair representing trophic level 2 (mean \pm sd = 1.8 ± 1.26 ‰, $n = 21$, Fig S1).”

Reviewer comments & author rebuttal 2:

I agree with the other reviewers that this was an impressive dataset, analysis, and compilation of data, however, that is also where it is deficient in so many ways. The authors argue for individual diets of bears such that Lines 114-117: "Populations range from tracking food resource pulses, such as spawning fish, scavenging on ungulate carcasses or preying on ungulate calves, or feeding extensively on invertebrates, to populations using primarily fruiting plant-based diets." With this breadth of diet, we would expect a simple analysis to document this which would include a figure like Figure S1 of all bears in their dataset. Presumably, bear foraging on carcasses would be different from those foraging on fruiting plants are at least separation of bear pairs that exhibit different overall diet selection.

There is an impressive wealth of diet studies demonstrating the breadth of brown bear diets. The aim of our study was to determine the sources of this variation within populations.

Figure 1B shows the phenotypic variation of trophic position in the population. Given that Scandinavian brown bears are omnivores – all bears are expected to forage on all of these different foods, however the proportional composition of foods on the individual level might vary, which is what we argue here based on Figure 1B. We have added a second Panel to Figure S1 showing the raw stable isotopic signatures of bear hair and bear foods (scaled by their appropriate discrimination factors).

The authors dismissed my comments on moose hair and basically stated that moose hair is equal in isotopes to moose meat or tissue consumed by bear. I realize that bear consume hair with the carcass but bear are not digesting and assimilating hair as they are moose tissue. As stated by the authors, bear and moose hair reflects a diet over time whereas tissue is within weeks of consumption and assimilation in the bear. It is not clear how moose hair can be a proxy for diet of moose?

We agree that one of the greatest challenges facing successful application of stable isotope applications in dietary studies is the proper choice of discrimination factors. The task becomes more challenging with omnivores that access both plant and meat in their diets. This situation is the result of a general lack of controlled captive studies where diet and tissue isotope values can be manipulated over long

periods and the general “quick fix” used by those who mechanically apply the +3.4‰ for $\Delta^{15}\text{N}$ regardless. “Solutions” to this dilemma have emerged in the form of R packages like *SIDER* but they are only as good as the metadata they are based on.

In our case, we were forced to use moose hair $\delta^{15}\text{N}$ as a proxy for moose in bear diets because moose meat (presumably representing the bulk of ingested tissue by bears) was unavailable. This was the approach used by Urton et al. (2005) in their boreal foodweb reconstruction involving wolves following suitable arithmetic correction to convert hair to meat and then meat to consumer tissue. However, most importantly, the recent review paper by Stephens et al (2023) really does represent the most current and in-depth analysis of this question. Importantly, they report in Supplementary Material that $\Delta^{15}\text{N}$ for a herbivore in a C3 system (ours) for hair is 4.0‰ and for muscle is 3.9‰. For vertebrate omnivores in a C3 system, $\Delta^{15}\text{N}$ for hair is 3.2‰ and for muscle is 3.2‰. This indicates that for $\delta^{15}\text{N}$ (the isotope we used for trophic level determination), it does not matter if we use bear hair or bear muscle for dietary reconstructions and for moose, there is no evidence for an isotopic difference between hair and meat. So, current literature suggests that our trophic analysis is relatively unchanged by variance in discrimination factors (again for $\delta^{15}\text{N}$ and NOT for $\delta^{13}\text{C}$) and that our conclusions stand.

Stephens, R.B., O. N. Shipley, R. J. Moll. 2023. Meta-analysis and critical review of trophic discrimination factors ($\Delta^{13}\text{C}$ and $\Delta^{15}\text{N}$): importance of tissue, trophic level, and diet source. *Functional Ecology*

Urton, E.J.M., and K.A. Hobson. 2005. Intrapopulation variation in gray wolf isotope ($\delta^{15}\text{N}$ and $\delta^{13}\text{C}$) profiles: implications for the ecology of individuals. *Oecologia*14:317-326.

3. On the matter of home range estimation from GPS & VHF data

Reviewer comments & author rebuttal 1:

Environmental Similarity: There are several clarifications needed with this data that I believe the authors should consider:

1. What is the sample size for these GPS and VHF collars that had >1000 GPS or VHF locations? It is difficult to assess if every bear that hair collected during capture was analyzed (so sample size reflected in first line of the Results) or if the Environmental similarity was based on a subsample of this assuming GPS/VHF collars failed, bears died, etc., thus a smaller sample size than what was originally collared and had hair collected. Line 431-432 is not adequate to inform the reader of the sample size that this entire section and analysis was based.
2. 95% kernel density estimator – This estimator is lacking considerable information considering it was done with both autocorrelated GPS data and likely not autocorrelated data with VHF telemetry. The program used for the kernel density estimator should be included as well as the smoothing option would be a minimum amount of information needed here. It would also be helpful to understand the reason for this choice if it was to “extract habitat composition in each bear’s lifetime home range.” Do the authors believe this is the best choice of an estimator considering it has been documented to be one of these least representative estimators of home range across many species?

To comment 1 and 2:

We see that line 431 must have added confusion.

Our dataset pulled together information from very different sources (hair sampling, genetics, movement data), and the final sample size (n = 71) was the number of individuals for which we had information on all of these aspects. We have added information on the R package used to calculate home ranges. We rephrased the sentence to, line 448:

“Here, we accounted for environmental similarity by extracting habitat composition in each bear’s lifetime home range (n = 71). We constructed home ranges using a 95% kernel density estimator (kernelUD function) in the R package adehabitatHR (Calenge, 2006) using the reference bandwidth as smoothing parameter (option “href”). Bears were monitored for a minimum of 1000 GPS locations (n = 47) or were located via VHF on at least 25 days (n = 24).”

Using a traditional kernel density estimator (KDE) on autocorrelated data is known to underestimate the sizes of home ranges (e.g., Noonan et al 2019 “A comprehensive analysis of autocorrelation and bias in home range estimation”). However, since we used this method across all individuals, they are all subjected to this same bias in home range size estimation. Our collar take a GPS position every At the time it was computationally more straight forward to do KDE than to fit home ranges with, e.g., akde, given the number of individuals in our study. With akde one has to fit multiple movement models to each individual and select the best one using AIC, after which one can obtain the utilization distribution. This can take up to an hour per individual.

We would prefer to keep the analysis as is, however, we are prepared to revise our home range estimator, if the reviewer insists.

3. Extracting home range composition for the “median” year if multiple years were monitored needs additional details. This might be appropriate for bears monitored 1-3 years but if monitored 5-10 years I question the reliability of the Corine Landcover Maps specifically if they are static and collected/released annually or in 5 year intervals? The lack of details on this data source makes this difficult to evaluate.

This is a valid point. The way the model is specified controls for environmental similarity with a covariance matrix and to our knowledge, this effect cannot be variable over time. Therefore, similar to a genetic pedigree, environmental similarity among pairwise home ranges is treated as a constant. The Corine Landcover Map was static but manually updated by us based on annual data of forest clearcutting by the Swedish Forestry Agency (Skogsstyrelsen). We manually recategorized mature forest stands to clearcuts in years when they were harvested; 9 years after harvest we recategorized them as young forests and 20 years after harvest as mature stands.

The real question is – how much habitat change do long-lived individuals experience during their lifetime and is it enough to affect the environmental similarity matrix (i.e. clearcutting changing their home range composition so substantially that its similarity relative to other home ranges changes).

We have addressed this comment in the following way – for individuals with > 4 monitoring years, we extracted home range composition in each year we monitored their trophic position. We plotted these annual values (points) alongside the value for the median year used in our analysis (line) and alongside the data distribution across all individuals. In fact, our annually updated maps reveal extremely little annual change in home range composition proportional to the overall home range size. We are therefore confident that the environmental similarity matrix is robust.

We have added this Figure to the supplementary material S5:

“Alternatively, resource abundance within home ranges could also change over the lifetime of individuals if habitat composition changes. Though our Corine Landcover Map was static, we accounted for forestry activity by manually updating the map with newly emerging clearcuts, based on annual data of forest clearcutting by the Swedish Forestry Agency (Skogsstyrelsen). We manually recategorized mature forest stands to clearcuts in years when they were harvested; 9 years after harvest we recategorized them as young forests and 20 years after harvest as mature stands. As environmental similarity is fitted with a constant environmental similarity matrix, we here provide a validation that the proportion of forest cover (FigS5B) and disturbed forest (FigS5C) in a bear’s home range only shifted minorly over the lifetime of an individual. We cannot exclude the possibility that we missed to account for unregistered changes in habitat composition, in particular forest disturbance.”

And referenced it in the main text line 458:

“Annual changes in home range habitat composition were negligible (Fig S5).”

Reviewer comments & author rebuttal 2:

Also, using KDE home ranges with reference bandwidth smoothing likely severely over-estimates the size of the area used by each bear. Even if we overlook the issue of GPS versus VHF telemetry in the sample size (Bears were monitored for a minimum of 1000 GPS locations ($n = 47$) or were located via VHF on at least 25 days ($n = 24$) so comparing home ranges of 1,000 to 25 locations as being equal, but centroids to core home ranges for home ranges that are considerably larger than what the bear uses makes this analyses difficult to interpret and justify. The authors do not provide this raw data, shapes of the home ranges used to determine overlap/distance between core areas, or how using 1000 versus 25 locations depending on bear would influence their results. Even in their rebuttal to my comments on this, a sentence is cut off mid way.

Thank you for your comment. In fact, KDE is known to underestimate home range sizes, not overestimate. To address your concern we have now re-estimated all home ranges using autocorrelated kernel density estimator (AKDE) from the ctmm package. We have extracted home range composition in these new home range polygons to assess the effect of environmental similarity and have extracted new home range centroids from which we calculated pairwise spatial distance (used in the supplementary material). Importantly, home range sizes using either estimator were strongly correlated with most home ranges having an estimated larger home range size while a few others were estimated at smaller range sizes (see Figure below). As accordingly centroids shifted, pairwise spatial distances tended to become longer for a few bears with significant changes in estimated home range size.

We have rerun all statistical models in the main text and supplement.

Updating the home range estimator did not significantly change our results – the variance explained by environmental similarity decreased from a median 9% to a median 5%, while the variance explained by spatial distance increased from a median 50% to a median 63%.

4. On the matter of code availability

Reviewer comments & author rebuttal 2:

The code is mostly output only code. Nobody with a comparable dataset could replicated their research and we are not even able to replicated what they assessed with so much data missing. There is no raw data to actually perform any analyses in a reproducible way. The authors saved .rds files after running analyses on raw data thus we can only evaluate what the authors chose to provide. They included code for reproducing figures that are already included in the manuscript which seems to have limited value.

The clean and complete dataset, i.e., without errors or missing data, that was generated and analyzed during this study and code to reproduce all analyses (including the supplementary analyses) is provided under the Open Science Framework: <https://doi.org/10.17605/OSF.IO/68B9U>. This is in line with Nature communications Data and Code sharing policies.

Providing raw data as rds files is common standard. This file format is advantageous over other data formats as it is more stable (files cannot be opened and cell contents changed, no problems with decimal delimiter). Our final dataset indeed includes a column of model derived data, i.e., maternal trophic position. Code to reproduce this column and to derive this final dataset is provided in the Supplementary Code S1-S3.

Given that we run Bayesian models, we saved model outputs to reproduce exact point estimates but all code to run models is provided. This is also standard practice.

As the code is annotated, someone that is interested to replicate the study with their own data should be able to adapt our code to their own data.

Thus, the above statement is incorrect.

Specifically, there are no bear telemetry locations to reproduce the home range estimates.

It is correct that we do not provide raw GPS data but instead a preprocessed data frame of spatial distances and environmental composition.

We further do not provide the data and code to create home ranges and extract spatial covariates, for two reasons:

a) The spatial layers are products that we bought ourselves and we do not have the right to distribute these.

b) We do not provide raw GPS & VHF data for our long-term study as we are currently not in the position to make this database entirely open access. All data are archived in the wireless remote animal monitoring server at SLU in Sweden and we would provide access to data and code upon reasonable request. As an example, we have now included the raw GPS data of one individual and a script how home ranges were estimated using ctm in the OSF code & data repository. We have also updated our Data and code availability statement to:

“Primary data and code to reproduce all analyses are provided in the OSF repository; <https://doi.org/10.17605/OSF.IO/68B9U> (Hertel, 2023). Restrictions apply to the GPS data and code to reproduce home ranges; these can be obtained from the first author upon reasonable request.”

They included bear C13 and N15 for 71 bears but some females had 1-7 daughters and appear to be considered independent or separate samples.

The model in the main text analysis N15 for 71 female bears with 213 records which are daughters with known information on their mother’s trophic position (n=33 mothers). We analyze stable isotopes for all bears in the Supplementary material S2 & S3 and both code and raw data are provided in the repository. In fact, the cleaned dataset used in the main analysis is built in the code “Supplement S1_S3.rmd”, it is therefore reproducible how we derived the final dataset from the total dataset.

Calenge C, 2006. The package “adehabitat” for the R software: a tool for the analysis of space and habitat use by animals. *Ecological modelling* 197:516-519.

Downs JA, Horner MW, 2008. Effects of Point Pattern Shape on Home-Range Estimates. *The Journal of Wildlife Management* 72:1813-1818. doi: <https://doi.org/10.2193/2007-454>.

Hertel AG, 2023. Data&Code: The ontogeny of individual specialization. doi: <https://doi.org/10.17605/OSF.IO/68B9U>.

Post DM, 2002. Using stable isotopes to estimate trophic position: models, methods, and assumptions. *Ecology* 83:703-718. doi: 10.1890/0012-9658(2002)083[0703:USITET]2.0.CO;2.